# Generalizable Representation Learning for fMRI-based Neurological Disorder Identification

**Wenhui Cui**                                                                    *wenhuicu@usc.edu*
*Ming Hsieh Department of Electrical and Computer Engineering*
*University of Southern California*

**Haleh Akrami**                                                                    *akrami@usc.edu*
*Ming Hsieh Department of Electrical and Computer Engineering*
*University of Southern California*

**Anand A. Joshi**                                                                  *ajoshi@usc.edu*
*Ming Hsieh Department of Electrical and Computer Engineering*
*University of Southern California*

**Richard M. Leahy**                                                                *leahy@usc.edu*
*Ming Hsieh Department of Electrical and Computer Engineering*
*University of Southern California*

**Reviewed on OpenReview:** *https://openreview.net/forum?id=zF9IrMTjCC*

## Abstract

Despite the impressive advances achieved using deep learning for functional brain activity analysis, the heterogeneity of functional patterns and the scarcity of imaging data still pose challenges in tasks such as identifying neurological disorders. For functional Magnetic Resonance Imaging (fMRI), while data may be abundantly available from healthy controls, clinical data is often scarce, especially for rare diseases, limiting the ability of models to identify clinically-relevant features. We overcome this limitation by introducing a novel representation learning strategy integrating meta-learning with self-supervised learning to improve the generalization from normal to clinical features. This approach enables generalization to challenging clinical tasks featuring scarce training data. We achieve this by leveraging self-supervised learning on the control dataset to focus on inherent features that are not limited to a particular supervised task and incorporating meta-learning to improve the generalization across domains. To explore the generalizability of the learned representations to unseen clinical applications, we apply the model to four distinct clinical datasets featuring scarce and heterogeneous data for neurological disorder classification. Results demonstrate the superiority of our representation learning strategy on diverse clinically-relevant tasks.

## 1 Introduction

Deep learning based approaches have demonstrated success in analyzing brain connectivity based on functional magnetic resonance imaging (fMRI) (Gadgil et al., 2020; Ahmedt-Aristizabal et al., 2021), but the scarcity and heterogeneity of fMRI data still pose challenges in clinical applications such as identifying neurological disorders. fMRI plays a vital role in identifying biomarkers for neurological disorders (Pitsik et al., 2023; Li et al., 2021). However, clinical fMRI datasets are not only high-dimensional and spatiotemporally complex but are also characterized by high variability among subjects and a limited number of data, posing significant challenges for training deep learning models to predict neurological disorders (Akrami et al., 2021). Medical datasets typically contain an abundance of healthy control data but often face a scarcity of clinical data collected for any particular neurological disorder. Simply training a model by aggregating

all healthy control and clinical data may cause limited generalization and bias due to the data imbalance and heterogeneity in clinical features. This can in-turn lead to poor performance in the group with clinical pathology (Azizi et al., 2023). Moreover, deep learning models often struggle to achieve satisfactory performance on small-scale clinical datasets, frequently under-performing compared to traditional machine learning methods (Akrami et al., 2024; 2021). Considering the limitations of both data and existing deep learning models, we explore representation learning on fMRIs, aiming to extract meaningful and generalizable features from data by learning inherent functional activity patterns. Driven by the need to learn generalizable representations, we leverage self-supervised learning and meta-learning as the foundation of our representation learning approach.

Self-supervised learning is popular in representation learning and has shown the ability to improve the generalization of features (Reed et al., 2022; You et al., 2020b). In contrast to fully-supervised tasks such as classification or segmentation, self-supervised tasks are typically designed to learn intrinsic features that are not specific to a particular task (Taleb et al., 2020). Contrastive self-supervised learning applied to fMRI classification has demonstrated the ability to prevent over-fitting on small medical datasets and address high intra-class variances (Wang et al., 2022). For our proposed approach, we apply contrastive self-supervised learning, known to be effective in representation learning (Azizi et al., 2023), to the healthy control data to learn more generalizable features.

Now that self-supervised learning is applied to learn generalizable representations from abundant healthy control data, we need an effective approach to transfer the learned knowledge to the scarce clinical data. For this, we adopt a meta-representation learning approach (Liu et al., 2020a), which leverages meta-learning to enhance generalization across domains. Meta-learning has recently gained tremendous attention because of its learning-to-learn mechanism, which strongly increases the generalizability of models across different tasks (Zhang et al., 2019; Liu et al., 2020a; Finn et al., 2017). By employing a bi-level optimization scheme (Finn et al., 2017), the model is trained to generalize effectively to unseen domains. Meta-learning is particularly effective in a low-data regime (Zhang et al., 2019), making it well-suited for clinical applications.

In this work, we introduce a novel representation learning strategy, Meta Transfer of Self-supervised Knowledge (MeTSK), which leverages meta-learning to generalize self-supervised features from large-scale control datasets (source domain) to scarce clinical datasets (target domain). MeTSK not only enables effective knowledge transfer from source to target domains but also enhances the model's ability to generalize to new and unseen clinical data in challenging applications by leveraging the learned generalization from control features to scarce clinical features. In summary, our contribution is three-fold:

- We are the first to propose a novel representation learning approach for fMRI data to achieve generalization across various challenging neurological disorder classification tasks with limited data;

- The proposed approach can serve as a reliable feature extractor for future challenging tasks with limited data, where deep learning methods typically fail.

- We address the heterogeneity and scarcity of clinical fMRI data through the integration of meta-learning and self-supervised learning.

Our experiments are designed to demonstrate, i). the improved knowledge transfer from source to target datasets, we use a neurological disorder classification task to evaluate the performance on the target domain when applying MeTSK in a knowledge transfer task setting. ii) The generalization of representations to unseen clinical datasets. We evaluate the MeTSK model pre-trained with a source and a target dataset on unseen clinical datasets. Direct training of deep learning models on these challenging clinical datasets performed poorly due to limited training data and the heterogeneity of features, also resulting in worse performance compared to simpler machine learning classifiers. So acquiring generalizable representations is crucial for these clinical datasets. According to Kumar et al. (2022), fine-tuning can distort good pre-trained features and degrade downstream performance under large distribution shifts. Here we explored linear probing for evaluating the generalization of representations on distinct neurological disorder classification tasks. As we show below, we are able to consistently achieve superior classification performance for diverse neurological disorder identification tasks compared to linear classification directly using traditional functional connectivity features as input.

## 2   Related Work

fMRI data are widely used for identifying neurological disorders such as Alzheimer's Disease (AD) (LaMontagne et al., 2019), epilepsy (Gullapalli, 2011), Parkinson's Disease (PD) (Badea et al., 2017), and Attention-Deficit/Hyperactivity Disorder (ADHD) (Bellec et al., 2017). These disorders cause atypical brain activity that can be characterized by analyzing fMRI data. In a traditional setting, features such as functional connectivity between brain regions (Van Den Heuvel & Pol, 2010) and Amplitude of Low-Frequency Fluctuation (ALFF) features (Zou et al., 2008), are extracted from raw fMRIs for statistical analysis or as inputs to machine learning classifiers to identify neurological biomarkers (Akrami et al., 2024). Given the inherent graph structure of fMRI data, where brain regions can be considered as nodes and functional connectivity measures as the edges, Graph Neural Networks (GNNs) (Li et al., 2021; Gadgil et al., 2020; Wang et al., 2022) have emerged as the predominant approaches in the literature for deep learning. GNNs usually take graph-structured data as input and perform graph convolution based on the neighboring relations (defined by edges) between graph nodes (Kipf & Welling, 2016). Li et al. (2021) proposed a GNN model with a novel pooling strategy for Autism Spectrum Disorder (ASD) classification; Zhang et al. (2023) applied a local-to-global GNN to ASD and AD classification, which uses a population graph to make use of inter-subject correlations in the dataset. In this work, we adopted a popular spatio-temporal GNN as our backbone model.

In fMRI analysis, heterogeneity and disparities across datasets pose significant challenges for the generalization of models. Most existing methods developed for fMRIs focus on adapting models between closely related domains, such as generalizing across imaging sites within the same dataset. These methods are often designed to address site-specific variations, such as differences in imaging protocols or scanner types (Li et al., 2020; Shi et al., 2021; Liu et al., 2023). Other approaches, such as fine-tuning (Raghu et al., 2019) and multi-task learning (Huang et al., 2020), attempt to adapt to target domains but often fail to learn transferable features due to domain discrepancies (Liu et al., 2020b; Raghu et al., 2019) and may distort the learning of target-domain specific features (Chen et al., 2019).

Despite these efforts, there is no existing work exploring the generalization across domains with fundamentally different characteristics, such as healthy control data and clinical data from patients with neurological disorders. To mitigate this gap, we explore a representation learning strategy that seeks to enhance generalization to unseen domains not available during training. This challenge, typical in real-world clinical applications (Akrami et al., 2024; Badea et al., 2017), where new datasets may vary significantly and are often scarce, makes traditional deep learning methods less applicable. Our approach also serves as a reliable feature extractor for small-scale datasets, thereby enabling robust neurological disorder identification.

## 3   Methods

In this section, we introduce our proposed strategy, MeTSK, which improves the generalization of self-supervised fMRI features from a control dataset to clinical datasets. The proposed network architecture consists of a feature extractor that learns generalizable features from both source (control) and target (clinical) domains, and source and target heads to learn domain-specific features for the source and target domain, respectively. The bi-level optimization strategy is applied to learn generalizable features using a Spatio-temporal Graph Convolutional Network (ST-GCN) (Gadgil et al., 2020) as the backbone model. After MeTSK is trained on the control and clinical datasets, its generalization capability will be evaluated on unseen clinical datasets using linear probing. The methodology of MeTSK as well as the representation learning pipeline are shown in Figure 1.

### 3.1   Feature Extractor: ST-GCN

We adopt a popular model for fMRI classification, ST-GCN (Gadgil et al., 2020), as the backbone architecture to extract graph representations from both spatial and temporal information. A graph convolution and a temporal convolution are performed in one ST-GCN module shown in Fig. 2, following the details in Gadgil et al. (2020). The feature extractor includes three ST-GCN modules. The target head and the source head share the same architecture, which consists of one ST-GCN module and one fully-connected layer.

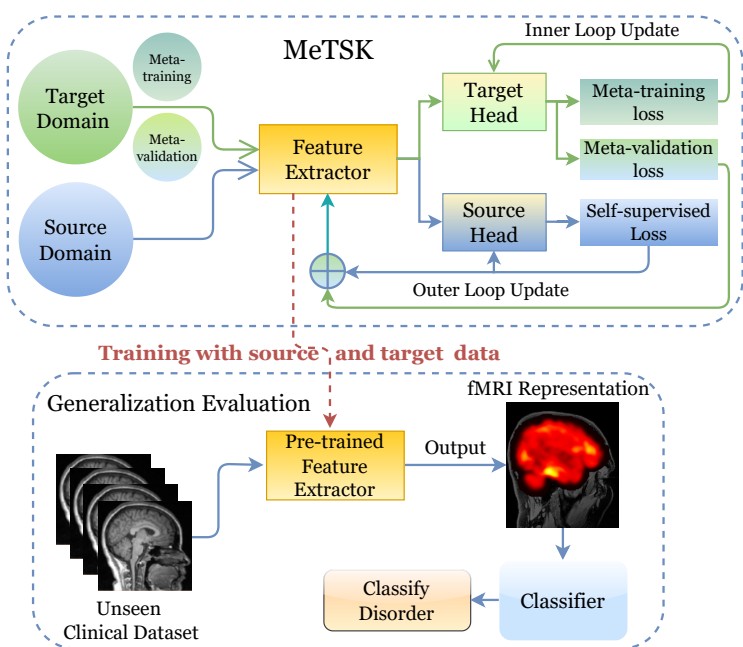

Figure 1: An illustration of MeTSK for generalizable representation learning. In MeTSK, two optimization loops are involved in training. The inner loop only updates the target head, while the outer loop updates the source head and feature extractor. The representation learning pipeline involves first training MeTSK on the source and target data, and then evaluate the learned representations on unseen clinical datasets using neurological disorder classification tasks.

To construct the graph, we treat brain regions parcellated by a brain atlas (Glasser et al., 2016) as the nodes and define edges using the functional connectivity between pairs of nodes measured by Pearson's correlation coefficient (Bellec et al., 2017). We randomly sample fixed-length sub-sequences from the whole fMRI time series to increase the size of training data by constructing multiple input graphs containing dynamic temporal information. For each time point in each node, a feature vector of dimension $C_i$ is learned. So for the $r$-th sub-sequence sample from the $n$-th subject, the input graph $X_i^{(n,r)}$ to the $i$-th layer has a dimension of $P \times L \times C_i$, where $P$ is the number of brain regions or parcels (nodes), $L$ is the length of the sampled sub-sequence, and $C_0 = 1$ for the initial input. In ST-GCN, a graph convolution (Kipf & Welling, 2016), applied to the spatial graph at time point $l$ in the $i$-th layer, can be expressed as follows.

$$X_{i+1}^{(n,r,l)} = D^{-1/2}(A + I)D^{-1/2}X_i^{(n,r,l)}W_{C_i \times C_{i+1}} \tag{1}$$

where $A$ is the adjacency matrix consisting of edge weights defined as Pearson's correlation coefficients, $I$ is the identity matrix, $D$ is a diagonal matrix such that $D_{ii} = \sum_j A_{ij} + 1$, and $W$ is a trainable weight matrix. We then apply 1D temporal convolution to the resulting sub-sequence of features on each node. A voting strategy is applied to combine predictions generated from different sub-sequences.

## 3.2 Bi-level Optimization

Assume there exists a source domain (healthy controls) $\mathcal{S}$ with abundant training data $X_{\mathcal{S}}$ and a target domain (clinical) $\mathcal{T}$, where the training data $X_{\mathcal{T}}$ is limited. A feature extractor $f(\phi)$, a target head $h_{\mathcal{T}}(\theta_t)$, and a source head $h_{\mathcal{S}}(\theta_s)$ are constructed to learn source features $h_{\mathcal{S}}(f(X_{\mathcal{S}}; \phi); \theta_s)$ as well as target features $h_{\mathcal{T}}(f(X_{\mathcal{T}}; \phi); \theta_t)$, where $\phi$, $\theta_t$, and $\theta_s$ are model parameters.

We introduce a bi-level optimization strategy to perform gradient-based update of model parameters (Finn et al., 2017; Liu et al., 2020a). The model first back-propagates the gradients through only the target head in several fast adaptation steps, and then back-propagates through the source head and feature extractor. Each step in a nested loop is summarized as follows:

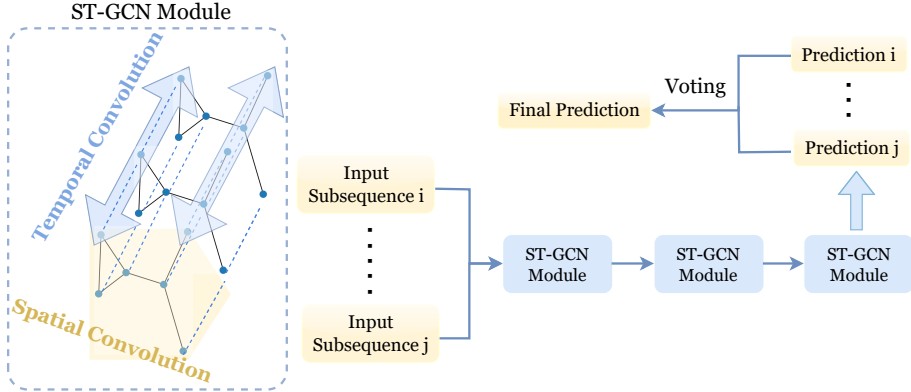

Figure 2: An illustration of the ST-GCN model architecture. Spatial graph convolution is first applied to the spatial graph at each time point. Then 1D temporal convolutions are performed along the resulting features on each node. Multiple sub-sequences are randomly sampled from the whole time series as input graphs.

**Outer loop** ($M$ iterations): Step 1. Initialize the target head and randomly sample target meta-training set $X_{\mathcal{T}_{tr}}$ and meta-validation set $X_{\mathcal{T}_{val}}$ from $X_{\mathcal{T}}$, where $X_{\mathcal{T}_{tr}} \bigcap X_{\mathcal{T}_{val}} = \emptyset$, $X_{\mathcal{T}_{tr}} \bigcup X_{\mathcal{T}_{val}} = X_{\mathcal{T}}$.

Step 2. **Inner loop** ($k$ update steps): Only target head parameters $\theta_t$ are updated using optimization objective $\mathcal{L}_{\mathcal{T}}$ (see below) for the target task. The parameter $\alpha$ is the inner loop learning rate, and $\theta_t^j$ is the target head parameter at the $j$-th update step.

$$\theta_t^{j+1} = \theta_t^j - \alpha \nabla_{\theta_t^j} \mathcal{L}_{\mathcal{T}}(h_{\mathcal{T}}(f(X_{\mathcal{T}_{tr}}; \phi^i); \theta_t^j)) \tag{2}$$

Step 3: After the inner loop is finished, freeze the target head and update feature extractor parameters $\phi$ and source head parameters $\theta_s$. The target loss $\mathcal{L}_{\mathcal{T}}$ and source loss $\mathcal{L}_{\mathcal{S}}$ are defined in the following section. The parameter $\beta$ is the outer loop learning rate, and $\lambda$ is a scaling coefficient.

$$\begin{aligned}
\{\theta_s^{i+1}, \phi^{i+1}\} = \{\theta_s^i, \phi^i\} - \beta(&\nabla_{\theta_s^i, \phi^i} \mathcal{L}_{\mathcal{S}}(h_{\mathcal{S}}(f(X_{\mathcal{S}}; \phi^i); \theta_s^i)) \\
&+ \nabla_{\phi^i} \lambda \mathcal{L}_{\mathcal{T}}(h_{\mathcal{T}}(f(X_{\mathcal{T}_{val}}; \phi^i); \theta_t^k)))
\end{aligned} \tag{3}$$

The target head, source head and feature extractor are updated in an alternating fashion. The target head is first trained on $X_{\mathcal{T}_{tr}}$ in the inner loop. In the outer loop, the feature extractor and source head are trained to minimize the generalization error of the target head on an unseen set $X_{\mathcal{T}_{val}}$ as well as to minimize the source loss. In this way, the feature extractor encodes features beneficial for both domains and the source head extracts features from the source domain that enable generalization to the target domain.

### 3.3 Contrastive Self-supervised Learning

To further boost the generalizability of features, we apply a graph contrastive loss (You et al., 2020a) to perform a self-supervised task on the source domain. We randomly sample sub-sequences $X^{(n,r_1)}$, $X^{(n,r_2)}$ ($r1 \neq r2$) from the whole fMRI time series for subject $n$ as the input graph features (Gadgil et al., 2020), which can be viewed as an augmentation of input graphs for ST-GCN. $X^{(n,r_1)}$ and $X^{(n,r_2)}$ should produce similar output graph features even though they contain different temporal information. The graph contrastive loss enforces similarity between graph features extracted from the same subject and dissimilarity between graph features extracted from different subjects (Chen et al., 2020b), so that the model learns invariant functional activity patterns across different time points for the same subject and recognizes inter-subject variances. A cosine similarity is applied to measure the similarity in the latent graph feature space (You et al., 2020a).

$$\mathcal{L}_{\mathcal{S}} = \frac{1}{N} \sum_{n=1}^{N} -\log \frac{\exp\left(sim(\tilde{X}_{\mathcal{S}}, n, n)/\tau\right)}{\sum_{m=1, m \neq n}^{N} \exp\left(sim(\tilde{X}_{\mathcal{S}}, n, m)/\tau\right)} \tag{4}$$

$$sim(X, n, m) = \frac{(X^{(n,r_1)})^\top X^{(m,r_2)}}{\|X^{(n,r_1)}\| \cdot \|X^{(m,r_2)}\|} \tag{5}$$

where $\tilde{X}_\mathcal{S} = h_\mathcal{S}(f(X_\mathcal{S}; \phi); \theta_s)$ is the generated graph representation, $\tau$ is a temperature hyper-parameter, and N is the total number of subjects in one training batch. By minimizing the graph contrastive loss on the source domain, the model produces consistent graph features for the same subject and divergent graph features across different subjects, which may be related to latent functional activities that reveal individual differences. Such intrinsic features are likely to be invariant across domains, promoting better generalization to unseen data.

The optimization objective $\mathcal{L}_\mathcal{T}$ of the target domain depends on the target task. In a classification task with class labels $Y_\mathcal{T}$, we adopt the Cross-Entropy loss. The total loss for the proposed MeTSK strategy is:

$$\mathcal{L}_{meta} = \mathcal{L}_\mathcal{S} + \lambda \mathcal{L}_\mathcal{T}$$
$$\mathcal{L}_\mathcal{T} = - \sum_{\text{classes}} Y_\mathcal{T} \log(h_\mathcal{T}(f(X_\mathcal{T}; \phi); \theta_t)) \tag{6}$$

### 3.4 Linear Probing

The bi-level optimization strategy and contrastive self-supervised learning work together to learn robust, domain-invariant features, enhancing the ability to generalize effectively to unseen clinical datasets. Here we evaluate the generalization of learned representations through linear probing, a crucial method for evaluating the quality of representations learned by the model (Chen et al., 2020a; Kumar et al., 2022). Linear probing involves freezing the parameters of a pre-trained model and training a linear classifier on the output. The intuition behind linear probing is that good representations should be linearly separable between classes (Chen et al., 2020a). We apply the MeTSK model pre-trained on the source and target domains to directly generate features for the unseen fMRI data without any fine-tuning. We then input these features into a linear classifier to perform neurological disorder classification. In alignment with common practices in the literature (Ortega Caro et al., 2023; He et al., 2020; Grill et al., 2020), we adopted linear SVM and logistic regression as probing classifiers. Both produce linear decision boundaries, enabling evaluation of representation quality through linear separability.

## 4 Datasets

We use HCP (Van Essen et al., 2013) as our source dataset due to its large size. For target datasets, we use the ADHD (Bellec et al., 2017) datasets, and the ABIDE dataset (Craddock et al., 2013) during the training of the MeTSK model. We then introduce four independent clinical datasets as held-out domains for evaluating generalization performance. For all the fMRI datasets used in this work, we parcellate the brain into 116 Regions of Interest (ROIs) using the Automated Anatomical Labeling (AAL) atlas in Tzourio-Mazoyer et al. (2002). The AAL atlas was defined based on brain anatomy. It divides the brain into 116 regions, including 90 cerebrum regions and 26 cerebellum regions. These 116 regions form the nodes of our graph. The fMRI data were reduced to a single time series per node by averaging the voxel-level signals within each region of interest (ROI). We then computed pairwise Pearson correlations between ROIs to obtain a functional connectivity matrix of size $116 \times 116$. The upper triangular part, including the diagonal, was extracted and flattened to form the input connectivity features for machine learning classifiers.

### 4.1 HCP Dataset

The healthy control data (source domain) is drawn from the Human Connectome Project (HCP) S1200 dataset (Van Essen et al., 2013). The HCP database includes 1,096 young adult (ages 22-35) subjects with resting-state-fMRI data collected at a total of 1200 time-points for each of four sessions. We used publicly available preprocessed fMRI data that follow the minimal preprocessing pipelines described in Glasser et al. (2013). Pre-processing steps include spatial distortion correction using field maps, motion correction through rigid-body registration, and EPI distortion correction. Functional images are registered to each subject's structural T1-weighted image, followed by spatial normalization to Montreal Neurological Institute (MNI) space. The data are then processed to align cortical data to a standard surface space (grayordinate system).

## 4.2 ADHD-Peking Dataset

The Attention-Deficit/Hyperactivity Disorder (ADHD-200) consortium data from the Peking site (Bellec et al., 2017) includes 245 subjects in total, with 102 ADHD subjects and 143 Typically Developed Controls (TDC). To model the situation where the clinical target data set is small, we use only the subset of the larger ADHD database that was collected from the Peking site. In the ablation studies, we explore the impact of using the complete set. We use the preprocessed data released on (`http://preprocessed-connectomes-project.org/adhd200/`). During preprocessing, the initial steps involve discarding the first four time points, followed by slice time and motion correction. The data is then registered to the MNI space, processed with a band-pass filter (0.009Hz - 0.08Hz), and smoothed using a 6 mm Full Width at Half Maximum (FWHM) Gaussian filter. All fMRIs have 231 time points after preprocessing.

## 4.3 ABIDE-UM Dataset

The Autism Brain Imaging Data Exchange I (ABIDE I) (Craddock et al., 2013) collects resting-state fMRI from 17 international sites. Similar to the ADHD dataset, we use only the subset of data from the UM site, which includes 66 subjects with Autism Spectrum Disorder (ASD) and 74 TDCs (113 males and 27 females aged between 8-29). We downloaded the data from `http://preprocessed-connectomes-project.org/abide/`, where data was pre-processed using the C-PAC pre-processing pipeline (Craddock et al., 2013). The fMRI data underwent several preprocessing steps: slice time correction, motion correction, and voxel intensity normalization. The data was then band-pass filtered (0.01–0.1 Hz) and spatially registered to the MNI template space using a nonlinear method. Spatial smoothing was applied using a 6mm FWHM Gaussian kernel to enhance the signal-to-noise ratio. All fMRIs have 296 time points.

## 4.4 Post-traumatic Epilepsy Dataset

We use the Maryland Traumatic Brain Injury (TBI) MagNeTs dataset (Gullapalli, 2011) for generalization performance evaluation. All subjects suffered a traumatic brain injury. Of these we used acute-phase (within 10 days of injury) resting-state fMRI from 36 subjects who went on to develop PTE and 36 who did not (Gullapalli, 2011; Zhou et al., 2012). The dataset was collected as a part of a prospective study that includes longitudinal imaging and behavioral data from TBI patients with Glasgow Coma Scores (GCS) in the range of 3-15 (mild to severe TBI). The individual or group-wise GCS, injury mechanisms, and clinical information is not shared. The fMRI data are available to download from FITBIR (`https://fitbir.nih.gov`). In this study, we used fMRI data acquired within 10 days after injury, and seizure information was recorded using follow-up appointment questionnaires. Exclusion criteria included a history of white matter disease or neurodegenerative disorders, including multiple sclerosis, Huntington's disease, Alzheimer's disease, Pick's disease, and a history of stroke or brain tumors. The imaging was performed on a 3T Siemens TIM Trio scanner (Siemens Medical Solutions, Erlangen, Germany) using a 12-channel receiver-only head coil. The age range for the epilepsy group was 19-65 years (yrs) and 18-70 yrs for the non-epilepsy group.

Pre-processing of the MagNeTs rs-fMRI data was performed using the BrainSuite fMRI Pipeline (BFP) (`https://brainsuite.org`). BFP is a software workflow that processes fMRI and T1-weighted MR data using a combination of software that includes BrainSuite, AFNI, FSL, and MATLAB scripts to produce processed fMRI data represented in a common grayordinate system that contains both cortical surface vertices and subcortical volume voxels (Glasser et al., 2013). The pre-processing follow the same minimal processing pipeline used for HCP data.

## 4.5 Alzheimer's Disease Dataset

Open Access Series of Imaging Studies (OASIS-3) (LaMontagne et al., 2019) is a longitudinal neuroimaging, clinical, and cognitive dataset for normal aging and Alzheimer Disease (AD), including 1379 participants: 755 cognitively normal adults and 622 individuals at various stages of cognitive decline ranging in age from 42-95 years old. Resting-state fMRI data was used for the classification of Alzheimer's Disease and is publicly available at `https://www.oasis-brains.org`. We randomly selected a subset of the OASIS-3 dataset, consisting of 42 subjects diagnosed as AD and 42 cognitively normal subjects for the downstream

clinical task. We are using a small subset to simulate the case where only limited disease data is available. There are 164 time points in each fMRI scan. The preprocessing steps used for OASIS-3 dataset are the same as used for the PTE dataset. We perform binary classification between AD and control subjects.

### 4.6 Parkinson's Disease Datasets

**TaoWu Dataset:** The Parkinson's Disease (PD) dataset collected by the group of Tao Wu (Badea et al., 2017) includes both T1-weighted and resting-state fMRI scans from 20 patients diagnosed with PD and 20 age-matched normal controls. All fMRI scans have 239 time points and were collected from a Siemens Magnetom Trio 3T scanner.

**Neurocon Dataset:** The Neurocon dataset is provided by the Neurology Department of the University Emergency Hospital Bucharest (Romania) (Badea et al., 2017), which includes 27 PD patients and 16 normal controls (with 2 replicate scans per subject). Both the rs-fMRI and T1-weighted scans were collected from a 1.5-Tesla Siemens Avanto MRI scanner. All fMRI scans have 137 time points. Both the Neurocon and TaoWU datasets were downloaded from `https://fcon_1000.projects.nitrc.org/indi/retro/parkinsons.html`. We pre-processed TaoWu and Neurocon fMRI data using the same preprocessing pipeline (BFP) that was applied to the PTE dataset.

## 5 Experiments and Results

### 5.1 Evaluation of Representation Transferability

To validate the effectiveness of MeTSK when training with a source and a target domain, we first evaluate the knowledge transfer from the HCP data (healthy controls) to ADHD-Peking data and ABIDE-UM data (clinical data). We designed experiments for tasks that perform ADHD v.s. TDC classification and ASD v.s TDC classification, respectively. We evaluate different strategies and compare their effectiveness in enhancing the knowledge transfer from a healthy dataset (source) to a clinical dataset (target).

For comparison, we designed (i) a baseline model using a ST-GCN with a supervised task directly trained on the target dataset (Baseline), (ii) a ST-GCN model fine-tuned on the target dataset after pre-training on HCP data (FT), (iii) a model performing multi-task learning on both source and target datasets simultaneously (MTL), and (iv) the proposed strategy, MeTSK. We incorporated MTL and FT methods for comparison in order to investigate whether MeTSK is superior to traditional approaches in terms of knowledge transfer. We compared several baseline methods: a Linear Support Vector Machine (SVM), a Random Forest Classifier (RF), a Multi-Layer Perceptron (MLP) consisting of three linear layers, an LSTM model for fMRI analysis (Gadgil et al., 2020), and a model combining a transformer and graph neural network (STAGIN) (Kim et al., 2021). For the SVM, RF, and MLP, the inputs are flattened functional connectivity features, calculated using the Pearson's correlation coefficient between fMRI time-series across pairs of brain regions defined in the AAL atlas. LSTM and STAGIN use raw fMRI time-series as their input.

We use 5-fold cross-validation to split training/testing sets on ADHD-Peking/ABIDE-UM data and use all HCP data for training. Model performance is evaluated using the average area-under-the-ROC-curve (AUC) as shown in Table 1. MeTSK achieved the best mean AUC of 0.6981 and 0.6967 for both target datasets, which is a significant improvement compared to the baseline model trained only on target data. MeTSK also surpassed the performance of fine-tuning and multi-task learning. The results demonstrate that the MeTSK strategy possesses the capability to enhance the knowledge transfer from healthy data to clinical data.

### 5.2 Evaluation of Representation Generalizability

Now we have a MeTSK model trained on HCP and ADHD-Peking data, we evaluate its generalization to four clinical datasets characterized by their small sample sizes and the inherent challenges they present in the identification of neurological disorders. The generalization performance was evaluated on challenging neurological disorder classification tasks. Specifically, the PTE dataset was used for classifying PTE subjects and non-PTE subjects; the OASIS-3 dataset was used for binary classification distinguishing Alzheimer's

Table 1: A comparison of mean AUCs of 5-fold cross-validation on ADHD dataset and ABIDE dataset using different methods: fine-tuning, multi-task learning, the proposed strategy MeTSK, and other baseline methods.

| Method | Source & Target | Target-only | ADHD-Peking | ABIDE-UM |
|---|---|---|---|---|
| SVM | ✗ | ✓ | $0.6182 \pm 0.0351$ | $0.6286 \pm 0.0635$ |
| RF | ✗ | ✓ | $0.6117 \pm 0.0503$ | $0.6266 \pm 0.0612$ |
| MLP | ✗ | ✓ | $0.6203 \pm 0.0468$ | $0.6312 \pm 0.0724$ |
| LSTM (Gadgil et al., 2020) | ✗ | ✓ | $0.5913 \pm 0.0510$ | $0.5936 \pm 0.0622$ |
| STAGIN (Kim et al., 2021) | ✗ | ✓ | $0.5638 \pm 0.0468$ | $0.5812 \pm 0.0684$ |
| ST-GCN (Baseline) | ✗ | ✓ | $0.6215 \pm 0.0435$ | $0.6051 \pm 0.0615$ |
| FT | ✓ | ✗ | $0.6213 \pm 0.0483$ | $0.6368 \pm 0.0454$ |
| MTL | ✓ | ✗ | $0.6518 \pm 0.0428$ | $0.6345 \pm 0.0663$ |
| **MeTSK (ours)** | ✓ | ✗ | $\mathbf{0.6981 \pm 0.0409}$ | $\mathbf{0.6967 \pm 0.0568}$ |

Disease (AD) from cognitively normal individuals; for the TaoWu and Neurocon datasets, we conducted binary classification to differentiate between Parkinson's Disease (PD) patients and healthy control subjects.

### 5.2.1 fMRI-based Foundation Models for Generalization Comparison

We compare our representation learning method with foundation models to assess how well our approach generalizes to unseen clinical data relative to these powerful, large-scale models. Foundation models are designed to capture generalizable representations across a wide variety of downstream tasks. By contrasting MeTSK's learned representations with those from foundation models, we can evaluate the effectiveness of our method in generating robust and generalizable features that perform well even with limited clinical data.

We compared our MeTSK model to a large pre-trained fMRI model, as detailed in Thomas et al. (2022). This model involves pre-training a Generative Pretrained Transformer (GPT) (Radford et al., 2019) on extensive datasets comprising 11,980 fMRI runs from 1,726 individuals across 34 datasets. During pre-training, the GPT model performs a self-supervised task to predict the next masked time point in the fMRI time-series. Their pre-trained model is publicly available at `https://github.com/athms/learning-from-brains`. We directly applied their pre-trained model to generate features. BrainLM (Ortega Caro et al., 2023), one of the latest foundation models developed for fMRI, was also incorporated into our experiments for comparison. BrainLM, consisting of a masked auto-encoder and a vision transformer, was trained on 6,700 hours of fMRI recordings. During pre-training, BrainLM incorporates a self-supervised task that predicts the randomly masked segments of time series in fMRI data, which is similar to the pre-training task in Thomas et al. (2022). Similarly, we directly applied their pre-trained model available at `https://github.com/vandijklab/BrainLM` to generate features. We also pre-trained a ST-GCN model on both HCP and ADHD-Peking datasets using only the proposed contrastive self-supervised learning (SSL). From this pre-trained SSL model, we again directly generated features for unseen clinical data.

### 5.2.2 Evaluation Results

We compare the linear probing performance across foundation model approaches as well as the performance of classifiers trained with functional connectivity features extracted from raw fMRI data. We employed machine learning classifiers, including a linear SVM, RF, and Logistic Regression (LR). The same 5-fold cross-validation was applied and AUCs for classification tasks were computed. Although complex classifiers could be used, the use of SVM, RF and LR isolates the quality of the learned representations from the model's complexity.

Our proposed MeTSK achieved the best performance on all four datasets, outperforming the two latest fMRI foundation models, as shown in Table 2. The features generated by MeTSK, the SSL model and BrainLM (Ortega Caro et al., 2023) all achieved better performance than functional connectivity features, owing to the knowledge learned from their extensive pre-training datasets. However, due to the domain gap between their pre-training data and the clinical datasets used here, the foundation models may require

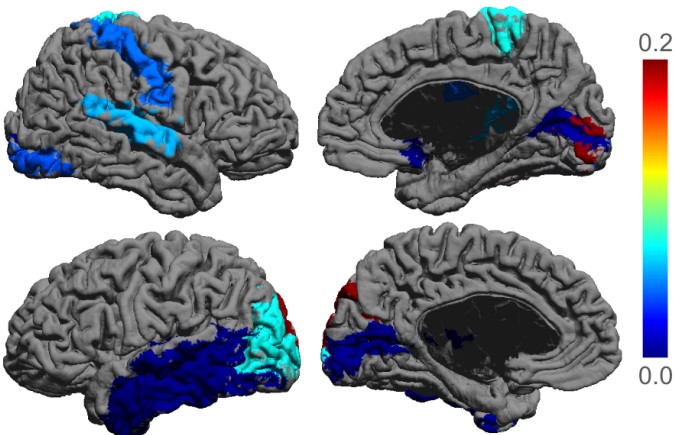

Figure 3: Feature importance map of PTE features generated from MeTSK shown as color-coded ROIs overlaid on the AAL atlas. The numbers represent the absolute value of coefficients from the trained SVM.

additional fine-tuning to further improve their performance. These results highlight MeTSK's potential in enhancing representation learning for clinical diagnostic purposes.

### 5.2.3 Interpretation of Learned Representations

To gain further insights and enhance the interpretability of the representations learned by MeTSK, we computed a feature importance map based on the positive coefficients from the SVM trained for PTE prediction. In a linear SVM, each feature within each ROI (brain region) is assigned a coefficient, indicating its importance in the model's decision-making process. The higher the absolute value of a coefficient, the greater its impact on the model's predictions. The coefficient for each ROI is calculated as the average of the coefficients for all features within that ROI. We extracted these coefficients for each ROI from the trained SVM and visualized them as a feature importance map overlaid on the brain, as shown in Fig. 3. Through observing the feature importance map, we identified that the most significant regions for PTE prediction are located in the temporal, parietal, and occipital lobes. Since epilepsy most commonly occurs in the temporal lobes, it is not surprising to see that they are among the regions identified from the feature important maps. A recent study on PTE (Akrami et al., 2024) also reported statistically significant differences between PTE and non-PTE groups in the parietal and occipital lobes. This alignment between our interpretation and clinical evidence demonstrate that the representations learned by MeTSK are not only predictive but also clinically meaningful.

### 5.3 Implementation Details

**Training:** To optimize model performance, we follow the training setting in (Gadgil et al., 2020) for the ST-GCN model. The length of input sub-sequences for ST-GCN is fixed at 128. We generate one meta-training batch by randomly selecting an equal number of samples from each class. The batch size is 32, both for the meta-training and the meta-validation set. We use an Adam optimizer (Kingma & Ba, 2014) with learning rate $\beta = 0.001$ in the outer loop, and an SGD optimizer (Ketkar, 2017) with learning rate $\alpha = 0.01$ in the inner loop. The number of inner loop update steps is 25. We set the hyper-parameter $\lambda = 30$ and the temperature parameter $\tau = 30$ to adjust the scale of losses following (Liu et al., 2020a; You et al., 2020a). Since contrastive loss converges slowly (Jaiswal et al., 2020), a warm-up phase is applied to train the model only on HCP data using the graph contrastive loss for the first 20 epochs. The number of total training epochs is 90. The hyper-parameters for the baseline ST-GCN model and the contrastive SSL loss follow the settings reported in Gadgil et al. (2020); You et al. (2020a). Other hyper-parameters in MeTSK were selected empirically by observing the convergence of the training loss. The hyper-parameters in SVM and RF during linear probing are selected via grid search with nested cross-validation. For the multi-task learning (MTL) implementation, we simply remove the inner loop in MeTSK and use all the training data

Table 2: Generalization evaluation results using 5-fold cross-validation: Mean and std of AUCs for PTE, AD, and PD classification using representations generated from different models (denoted as "linear probing" in the Table) as well as directly from functional connectivity features. The best performance achieved on each dataset is highlighted in bold.

|  |  | Linear Probing | | | | Connectivity Features |
|---|---|---|---|---|---|---|
|  |  | MeTSK | SSL | Thomas et al. (2022) | Ortega Caro et al. (2023) |  |
| PTE | SVM | **0.6415 ± 0.0312** | 0.5972 ± 0.0492 | 0.5369 ± 0.0451 | 0.6011 ± 0.0465 | 0.5697 ± 0.0477 |
|  | RF | 0.5392 ± 0.0553 | 0.5253 ± 0.0486 | 0.4814 ± 0.0664 | 0.5589 ± 0.0611 | 0.5081 ± 0.0612 |
|  | LR | 0.5770 ± 0.0635 | 0.5087 ± 0.0455 | 0.5199 ± 0.0678 | 0.5378 ± 0.0663 | 0.5190 ± 0.0566 |
| OASIS-3 | SVM | 0.6407 ± 0.0741 | 0.5992 ± 0.0867 | 0.6115 ± 0.0582 | 0.6028 ± 0.0831 | 0.5541 ± 0.0677 |
|  | RF | **0.6750 ± 0.0753** | 0.6055 ± 0.0682 | 0.6233 ± 0.0672 | 0.5604 ± 0.0607 | 0.5329 ± 0.0721 |
|  | LR | 0.5894 ± 0.0544 | 0.5044 ± 0.0581 | 0.5543 ± 0.0439 | 0.5207 ± 0.0437 | 0.5482 ± 0.0520 |
| Taowu | SVM | **0.6831 ± 0.1431** | 0.6371 ± 0.1578 | 0.5528 ± 0.1586 | 0.4937 ± 0.1506 | 0.5725 ± 0.1502 |
|  | RF | 0.6553 ± 0.1701 | 0.6273 ± 0.1679 | 0.4843 ± 0.1885 | 0.4891 ± 0.1710 | 0.6031 ± 0.1631 |
|  | LR | 0.5725 ± 0.1696 | 0.5100 ± 0.1674 | 0.5175 ± 0.1848 | 0.5387 ± 0.1985 | 0.5275 ± 0.2057 |
| Neurocon | SVM | **0.7529 ± 0.1579** | 0.5643 ± 0.1068 | 0.6433 ± 0.1535 | 0.6476 ± 0.1686 | 0.6599 ± 0.1807 |
|  | RF | 0.6230 ± 0.1658 | 0.5219 ± 0.1627 | 0.5813 ± 0.1818 | 0.6774 ± 0.1676 | 0.5427 ± 0.1715 |
|  | LR | 0.5885 ± 0.1815 | 0.5033 ± 0.1768 | 0.5238 ± 0.2001 | 0.5277 ± 0.1911 | 0.5163 ± 0.2167 |

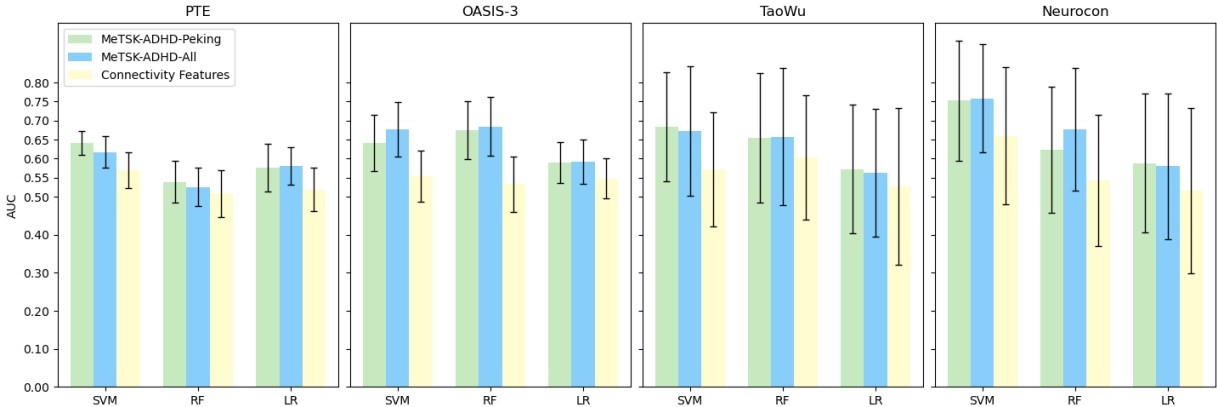

Figure 4: A comparison of the linear probing performance between MeTSK model trained with a single target site (MeTSK-ADHD-Peking, results from Table 2) and MeTSK model trained using all the target sites from ADHD-200 dataset (MeTSK-ADHD-All). Also shown are results from linear classifiers trained directly using connectivity features for baseline comparison. The height of each bar indicates the average AUC computed from a 5-fold cross-validation, while the error bars denote the standard deviations. MeTSK-ADHD-All achieved similar performance to MeTSK-ADHD-Peking on all the four downstream datasets.

to update the target head. Both heads and the feature extractor are updated simultaneously in one loop. For meta-learning, the target training set in each fold is further divided into a meta-training set $X_{\mathcal{T}_{tr}}$ of 157 subjects and a meta-validation set $X_{\mathcal{T}_{val}}$ of 39 subjects. Both the meta-train and meta-validation sets are randomly re-sampled in every iteration, and the target head is randomly initialized. In the training of MeTSK, binary classification is performed using one fully-connected layer followed by a sigmoid activation. The classification loss is the binary cross-entropy loss. .

**Evaluation:** For SSL and MeTSK, we use the pre-trained feature extractor for generating features. The generated features are graph-level representations, having a two dimensional feature matrix at each node (brain region). We averaged the features along the first dimension and applied Pinciple Component Analysis (PCA) to reduce the dimensionality before feeding the features into classifiers. The logistic regression used in the downstream experiments consists of a linear layer and followed by sigmoid activation. The SSL model trained on both HCP and ADHD-Peking data used the same contrastive loss and hyper-parameters in MeTSK. In our comparative analysis with a foundation model for fMRI (Thomas et al., 2022), we flatten the brain signals at each time-point and input the whole time-series without masking into the pre-trained

GPT model. This generates a feature embedding for each time-point. Since there is no class token in the pre-trained model, we averaged the feature embeddings across time-points (tokens) to get the output features for downstream datasets. We follow the other detailed settings of the pre-trained GPT model in (Thomas et al., 2022). For another foundation model BrainLM (Ortega Caro et al., 2023), we followed the instructions provided on `https://github.com/vandijklab/BrainLM/tree/main`. Specifically, we directly input the fMRI time-series into the pre-trained model without masking. Then we extracted the output class token as the generated features for clinical datasets.

**Computational Cost:** MeTSK introduces additional training time because there is a nested optimization loop within each training iteration. Training MeTSK takes approximately 6 hours on a single NVIDIA V100 GPU with 32 GB of memory. This nested gradient computation in the inner loop increases both training time and memory consumption. Larger GPU memory is recommended when scaling MeTSK to larger datasets or models. The SSL model requires about 4 hours of training using the same GPU. Baseline models such as ST-GCN, LSTM, and STAGIN typically take between 1.5 to 2 hours to train. Since we evaluate the two foundation models using pre-trained checkpoints, their inference cost is negligible. While MeTSK has higher training cost, it is a one-time cost at the pretraining stage. Once trained, the representations can be reused across downstream tasks with minimal computational overhead using linear probing.

## 6 Ablation Study

### 6.1 Training with More Clinical Data

In this paper we have explored a strategy MeTSK which trains a model to generalize from abundant normal features to scarce clinical features. To model this situation, we employed only a single site (Peking) from the ADHD-200 dataset to train our model. MeTSK emphasizes the model's ability to extrapolate from abundant healthy control features to scarce clinical features, equipping it with generalization capabilities for unseen real-world applications where clinical data may also be limited.

However, while MeTSK is intentionally tailored for training with small-scale clinical data, we also explored the impact of using more clinical target data to train the model. We used the data from all of the sites in the ADHD-200 datasets for training. The entire dataset comprised a total of 362 ADHD subjects and 585 TDCs, all pre-processed using the same steps as described in Section 4.1.2. The same linear probing was performed to evaluate the models on the four clinical datasets: PTE, OASIS-3, TaoWu, and Neurocon. As shown in Fig 4, MeTSK consistently demonstrated similar performance for various downstream tasks when trained using the larger clinical dataset. We computed the p-value using a paired Student's t-test, and showed that there is no significant difference between the downstream performance of the model trained with Peking site only (MeTSK-ADHD-Peking) and the model trained with the entire ADHD-200 dataset (MeTSK-ADHD-All). This underscores the model's robustness and its capacity to achieve effective generalization without relying on a large amount of training data. In some cases, training with the entire ADHD dataset achieved worse performance. The ADHD-All dataset includes fMRI scans from multiple imaging sites with different acquisition protocols, which can introduce distribution shifts. This phenomenon has been studied in prior work (the data addition dilemma) Shen et al. (2024), which highlights that adding more data from heterogeneous sources can in some cases degrade model performance due to domain shifts.

### 6.2 Ablation Study of MeTSK

We examine the individual contributions of self-supervised learning and meta-learning to the model performance on both target clinical datasets (ADHD-Peking, ABIDE-UM) in this section. To explore the effect of meta-learning, we designed an experiment using only the target (clinical) dataset in meta-learning (MeL). This approach involves removing the source head and the source loss during bi-level optimization. The target head is first trained on the ADHD/ABIDE meta-training set in the inner loop, followed by feature extractor learning to generalize on a held-out validation set in the outer loop. Our results, as shown in the last two rows of Table 3, reveal that the mean AUC improved from the baseline performance of 0.6215 to 0.6562 for ADHD classification, and from 0.6051 to 0.6675 for ASD classification without source domain knowledge.

Table 3: Ablation study on ADHD-Peking and ABIDE-UM dataset. The FT, MTL, and MeTSK methods are compared for two cases - transferring features from (i) a self-supervised source task and (ii) a sex classification source task, respectively. The last two rows are models trained only on target clinical data: a meta-learning model without source task and a baseline model ST-GCN.

| Dataset | ADHD-Peking | | ABIDE-UM | |
|---|---|---|---|---|
| Source Task | Self-supervision | Sex Classification | Self-supervision | Sex Classification |
| FT | $0.6213 \pm 0.0483$ | $0.6150 \pm 0.0497$ | $0.6368 \pm 0.0454$ | $0.6071 \pm 0.0742$ |
| MTL | $0.6518 \pm 0.0428$ | $0.6377 \pm 0.0512$ | $0.6345 \pm 0.0663$ | $0.6240 \pm 0.0711$ |
| **MeTSK** | $0.6981 \pm 0.0409$ | $0.6732 \pm 0.0579$ | $0.6967 \pm 0.0568$ | $0.6786 \pm 0.0749$ |
| MeL | $0.6562 \pm 0.0489$ | | $0.6675 \pm 0.0505$ | |
| ST-GCN (Baseline) | $0.6215 \pm 0.0435$ | | $0.6051 \pm 0.0615$ | |

To assess the contribution of self-supervised learning, we compared the impact of using a self-supervised task versus a sex classification task on the HCP dataset. Fine-tuning, multi-task learning, and MeTSK were implemented using sex classification (female vs male) as the source task. The same 5-fold cross-validation method was applied to compare the average AUC. As detailed in Table 3, all three methods: FT, MTL, and MeTSK, showed a degraded performance when transferring knowledge from the sex classification task. This suggests that the sex-related features of the brain may be less relevant to ADHD/ASD classification, negatively affecting the model's performance.

# 7 Discussion

Our proposed method, MeTSK, addresses the critical need for generalizable representations in clinical fMRI applications, particularly in scenarios with data scarcity and heterogeneity. Traditional deep learning approaches often struggle in these scenarios due to the lack of sufficient labeled data and high variability across subjects, as shown in our results (Appendix A.1) where simple machine learning classifiers, like SVM and RF, outperformed deep learning models when trained directly on limited clinical data. The high-dimensional nature and complex spatial-temporal dynamics of fMRI data make it even more challenging to extract diagnostic features given limited training data. These challenges underscore the importance of developing a generalizable representation learning strategy for clinical applications.

In this work, we primarily focus on evaluating representation generalization in low-data regimes. Although some datasets, such as OASIS-3, contain a large number of subjects, we only used a subset of the entire dataset and applied a 5-fold cross-validation protocol to maintain consistency across all datasets. Cross-validation provides a balanced estimate of generalization performance across multiple folds and ensures the evaluation setup is consistent across datasets, including those with naturally limited sample sizes. Although training on a small subset and evaluating on the entire dataset is interesting to explore, it remains challenging because the limited training subjects may not capture the full variability of the entire population.

Further explorations exist for expanding this approach in future work. One direction is to experiment with alternative functional brain atlases, such as DiFuMO (Dadi et al., 2020), or to incorporate different functional connectivity measures (e.g. Partial correlation) beyond Pearson's correlation (Varoquaux & Craddock, 2013; Smith et al., 2011). Note that our current approach is compatible with various choices of atlases and connectivity measures, which opens the potential to extend MeTSK to more advanced connectivity metrics and to easily adapt it to different atlas configurations. Additionally, exploring multi-class classification for distinguishing between multiple neurological disorders in downstream tasks rather than a simple binary classification could potentially provide further insights.

# 8 Conclusion

To summarize, we developed a representation learning strategy for neurological disorders, which achieved superior generalization to diverse small-scale clinical datasets and significantly improved the performance on various challenging neurological disorder classification tasks. By employing meta-learning, we enhance

the generalization capabilities of self-supervised features from the source domain (control) to target clinical domains, ensuring that the learned representations capture intrinsic patterns in functional brain activity and such patterns are shared across datasets. The learned representations enable effective classification even with simple linear classifiers. This demonstrates the quality of the learned features and their generalizability across unseen clinical datasets.

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

Table 4: Mean and std of AUCs of classification results using 5-fold cross-validation

| Methods | PTE | OASIS-3 | TaoWu | Neurocon |
|---|---|---|---|---|
| SVM | $0.5697 \pm 0.0477$ | $0.5541 \pm 0.0677$ | $0.5725 \pm 0.1502$ | $0.6599 \pm 0.1807$ |
| RF | $0.5081 \pm 0.0612$ | $0.5329 \pm 0.0721$ | $0.6031 \pm 0.1631$ | $0.5427 \pm 0.1715$ |
| MLP | $0.5111 \pm 0.0402$ | $0.5231 \pm 0.0689$ | $0.5875 \pm 0.1631$ | $0.5313 \pm 0.1721$ |
| ST-GCN (Gadgil et al., 2020) | $0.5108 \pm 0.0718$ | $0.5079 \pm 0.0833$ | $0.5377 \pm 0.1739$ | $0.5089 \pm 0.1890$ |
| LSTM (Gadgil et al., 2020) | $0.5031 \pm 0.0712$ | $0.4675 \pm 0.0880$ | $0.5019 \pm 0.1721$ | $0.4914 \pm 0.1799$ |
| STAGIN (Kim et al., 2021) | $0.4517 \pm 0.0806$ | $0.4891 \pm 0.0904$ | $0.4332 \pm 0.1859$ | $0.4470 \pm 0.1822$ |

Xi Sheryl Zhang, Fengyi Tang, Hiroko H Dodge, Jiayu Zhou, and Fei Wang. Metapred: Meta-learning for clinical risk prediction with limited patient electronic health records. In *Proceedings of the 25th ACM SIGKDD International Conference on Knowledge Discovery & Data Mining*, pp. 2487–2495, 2019.

Yongxia Zhou, Michael P Milham, Yvonne W Lui, Laura Miles, Joseph Reaume, Daniel K Sodickson, Robert I Grossman, and Yulin Ge. Default-mode network disruption in mild traumatic brain injury. *Radiology*, 265 (3):882, 2012.

Qi-Hong Zou, Chao-Zhe Zhu, Yihong Yang, Xi-Nian Zuo, Xiang-Yu Long, Qing-Jiu Cao, Yu-Feng Wang, and Yu-Feng Zang. An improved approach to detection of amplitude of low-frequency fluctuation (alff) for resting-state fmri: fractional alff. *Journal of neuroscience methods*, 172(1):137–141, 2008.

# A  Appendix

## A.1  Applying Deep Learning Models to Small-Scale Clinical Datasets

We present the classification results on the clinical datasets mentioned in Section 5.2 using some popular deep learning models. These models demonstrated severe over-fitting and poor generalization, achieving lower AUCs than traditional machine learning classifiers. The deep learning models struggled to learn from the limited and highly variable clinical data, where the high-dimensional and complex nature of fMRI data and small sample sizes amplified the over-fitting issues. These results shown in Table 4 emphasized our motivation for developing a representation learning approach capable of generating generalizable features for such challenging datasets.

Due to methodological and experiment setting differences across previous works (e.g., varying cross-validation splits, fMRI atlas choices, and number of training samples used), direct numerical comparisons should be interpreted cautiously. For example, different fMRI atlases define varying numbers of ROIs, which directly impact the dimensionality of the input connectivity features and change the number of nodes in the input graphs. We include these published results from previous works to contextualize our results within the broader literature, as shown in Table 5. Unlike previous works that often leverage larger datasets, our experimental design simulates real-world clinical scenarios with limited training data. This allows us to assess the robustness and generalization capability of MeTSK under data-scarce conditions—settings that are common in clinical practice.

## A.2  P-values Computed from Wilconxon Signed-Rank Tests

We report the p-values computed using the Wilcoxon signed-rank test between the results from Connectivity Features, MeTSK-ADHD-All, and MeTSK-ADHD-Peking in Table 6. As shown, the comparisons between MeTSK-ADHD-Peking and MeTSK-ADHD-All consistently yield high p-values, indicating no statistically significant difference between them. In contrast, comparisons between MeTSK-ADHD-Peking and Connectivity Features result in smaller p-values, suggesting more substantial differences in performance.

Table 5: Summary of reported results on our four clinical datasets in the literature as well as the best results achieved by MeTSK. Note the significant differences in metrics, cross-validation splits, training size, and fMRI atlases.

| Dataset | Method | Results | Metric | CV Splits | # Training Samples | fMRI Atlas | # ROIs |
|---|---|---|---|---|---|---|---|
| Neurocon | Xu et al. (2024) | 0.75 | Accuracy | 10-fold CV | 42 | Schaefer (Schaefer et al., 2018) | 100 |
| | Shi et al. (2022) | 0.667 | AUC | 10-fold CV | 42 | Brainnetome (Fan et al., 2016) | 246 |
| | Vigneshwaran et al. (2024) | 0.5 | F1 Score | 40% for testing | 42 | Harvard-Oxford | 69 |
| | Li et al. (2021) | 0.66 | Accuracy | 10-fold CV | 42 | Schaefer | 100 |
| | **MeTSK (Ours)** | 0.7529 | AUC | 5-fold CV | 42 | AAL | 116 |
| Taowu | Xu et al. (2024) | 0.775 | Accuracy | 10-fold CV | 40 | Schaefer | 100 |
| | Vigneshwaran et al. (2024) | 0.6 | F1 Score | 40% for testing | 40 | Harvard-Oxford Atlas (Wu et al., 2018) | 69 |
| | Li et al. (2021) | 0.675 | Accuracy | 10-fold CV | 40 | Schaefer | 100 |
| | **MeTSK (Ours)** | 0.6831 | AUC | 5-fold CV | 40 | AAL | 116 |
| OASIS-3 | Han et al. (2024) (Proposed GCN) | 0.800 | Accuracy | 10-fold CV | 1006 | Schaefer | 100 |
| | Han et al. (2024) (Baseline GCN) | 0.686 | Accuracy | 10-fold CV | 1006 | Schaefer | 100 |
| | Han et al. (2024) (SVM) | 0.568 | Accuracy | 10-fold CV | 1006 | Schaefer | 100 |
| | **MeTSK (Ours)** | 0.675 | AUC | 5-fold CV | 84 | AAL | 116 |
| PTE | Akrami et al. (2024) (KSVM) | 0.6100 | AUC | Leave-one-out CV | 72 | USCLobes (Joshi et al., 2022) | 16 |
| | Akrami et al. (2024) (GCN) | 0.5900 | AUC | Leave-one-out CV | 72 | USCLobes | 16 |
| | **MeTSK (Ours)** | 0.6415 | AUC | 5-fold CV | 72 | AAL | 116 |

Table 6: Wilcoxon signed-rank test p-values between MeTSK and two other methods in Figure 4 across all datasets and classifiers. Columns represent comparisons against MeTSK-ADHD-Peking using MeTSK-ADHD-All and Connectivity Features, respectively. A higher p-value indicates a less significant difference between methods.

| Dataset | Classifier | vs ADHD-All | vs Connectivity Features |
|---|---|---|---|
| PTE | SVM | 0.4375 | 0.3125 |
| | RF | 0.8125 | 0.4375 |
| | LR | 1.0 | 0.3125 |
| OASIS-3 | SVM | 0.625 | 0.1250 |
| | RF | 0.8125 | 0.1250 |
| | LR | 0.8125 | 0.3125 |
| TaoWu | SVM | 0.8125 | 0.1250 |
| | RF | 0.8125 | 0.3125 |
| | LR | 0.625 | 0.3125 |
| Neurocon | SVM | 0.8125 | 0.3125 |
| | RF | 0.4375 | 0.3125 |
| | LR | 0.675 | 0.3125 |

## A.3 Convergence Analysis of MeTSK

To monitor the convergence and stability of MeTSK training, we track both the self-supervised loss and the cross-entropy loss computed from the frozen target head's predictions using the meta-validation set in Step 3 (refer to Method Section 3.2). We plot the evolution of the two losses over training iterations in Figure 5. During training, we observe that the self-supervised loss decreases well, while the meta-validation cross-entropy (classification) loss also shows a decreasing trend overall, although with some fluctuations. Fluctuations are expected, as the loss is computed on a small (39-subject) randomly sampled meta-validation set. Both the meta-train and meta-validation sets are re-sampled in each training iteration, and the target classification head is randomly initialized, introducing further variability in the observed loss values. This indicates that the model is able to jointly optimize the self-supervised and classification losses in Step 3, and that the convergence is not negatively affected by the frozen target head trained in Step 2.

The classification loss on the meta-training set after inner-loop updates in Step 2 consistently decreases to a stable range across training epochs, as shown in Figure 5. Additionally, the final meta-training classification accuracies after inner-loop updates fall within a stable range of 72% to 81%. This suggests that the target head can quickly adapt to the updated features and the optimization steps in Step 3 do not negatively affect the convergence of the inner-loop updates in Step 2.

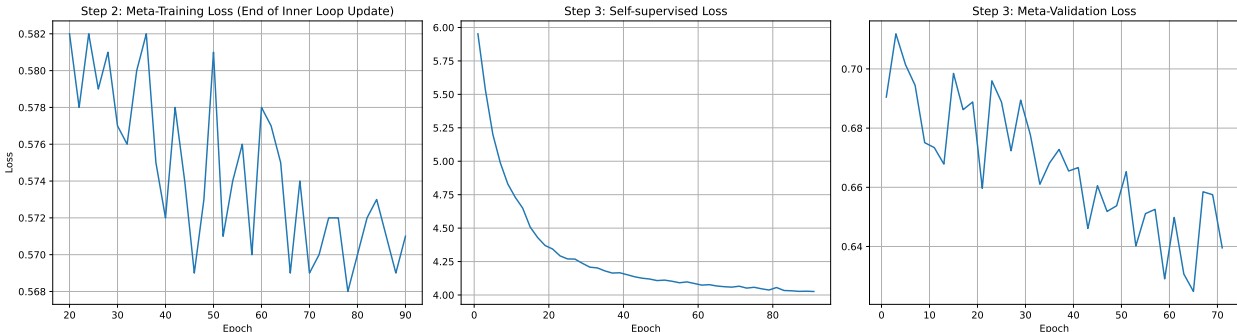

Figure 5: Loss evolution during MeTSK training over 90 epochs. The first 20 epochs is a warm-up phase where only the self-supervised task is trained. All loss values are averaged over batches and recorded every 2 epochs. **Left**: Classification loss on the meta-training set after the inner-loop updates in Step 2. We record the loss at the last update step in each training iteration. This loss remains within a stable range and gradually decreases, indicating that the target head effectively adapts using features trained in Step 3 without negative interference. **Middle**: Self-supervised loss from Step 3, showing a steady decline. **Right**: Meta-validation classification loss in Step 3, computed from the frozen target head, exhibits a generally decreasing trend with some fluctuations.

Table 7: Comparison of linear probing performance from foundation models Thomas et al. (2022); Ortega Caro et al. (2023) and their fine-tuned versions on four clinical datasets using 5-fold cross-validation. We directly used results of the foundation model w/o finetuning from Table 2 for comparison. Mean ± std AUCs are reported for both pre-trained and fine-tuned models. Fine-tuned results are obtained by further training the foundation models on the ADHD-Peking dataset.

| Dataset | Classifier | Thomas et al. (2022) | Fine-tuned | Ortega Caro et al. (2023) | Fine-tuned |
|---------|-----------|----------------------|------------|----------------------------|------------|
| PTE | SVM | $0.5369 \pm 0.0451$ | $0.5291 \pm 0.0473$ | $0.6011 \pm 0.0465$ | $0.6102 \pm 0.0498$ |
| | RF | $0.4814 \pm 0.0664$ | $0.4795 \pm 0.0651$ | $0.5589 \pm 0.0611$ | $0.5502 \pm 0.0584$ |
| | LR | $0.5199 \pm 0.0678$ | $0.5252 \pm 0.0662$ | $0.5378 \pm 0.0663$ | $0.5415 \pm 0.0691$ |
| OASIS-3 | SVM | $0.6115 \pm 0.0582$ | $0.6152 \pm 0.0603$ | $0.6028 \pm 0.0831$ | $0.5955 \pm 0.0792$ |
| | RF | $0.6233 \pm 0.0672$ | $0.6184 \pm 0.0660$ | $0.5604 \pm 0.0607$ | $0.5601 \pm 0.0622$ |
| | LR | $0.5543 \pm 0.0439$ | $0.5482 \pm 0.0455$ | $0.5207 \pm 0.0437$ | $0.5280 \pm 0.0411$ |
| TaoWu | SVM | $0.5528 \pm 0.1586$ | $0.5477 \pm 0.1602$ | $0.4937 \pm 0.1596$ | $0.4913 \pm 0.1530$ |
| | RF | $0.4843 \pm 0.1885$ | $0.4782 \pm 0.1903$ | $0.4891 \pm 0.1710$ | $0.4965 \pm 0.1684$ |
| | LR | $0.5175 \pm 0.1848$ | $0.5083 \pm 0.1822$ | $0.5387 \pm 0.1985$ | $0.5280 \pm 0.1971$ |
| Neurocon | SVM | $0.6433 \pm 0.1535$ | $0.6382 \pm 0.1510$ | $0.6476 \pm 0.1686$ | $0.6488 \pm 0.1755$ |
| | RF | $0.5813 \pm 0.1818$ | $0.5755 \pm 0.1799$ | $0.6774 \pm 0.1676$ | $0.6703 \pm 0.1728$ |
| | LR | $0.5238 \pm 0.2001$ | $0.5104 \pm 0.2153$ | $0.5277 \pm 0.1911$ | $0.5292 \pm 0.1984$ |

## A.4 Linear Probing Results of Fine-tuned Foundation Models

To ensure a fair comparison, we fine-tuned the baseline foundation models on the same dataset—HCP combined with ADHD-Peking—that was used for training MeTSK. Notably, the pre-training datasets for these foundation models already included the HCP data, so aligning the training data required only fine-tuning on the ADHD-Peking dataset. Specifically, we load the pre-trained foundation models and fine-tune them on the ADHD-Peking dataset using the same self-supervised pre-training objective originally used to train the models—reconstructing masked time points. After fine-tuning on ADHD-Peking, we performed linear probing on the four clinical datasets. Our results in Table 7 show that the linear probing performance of the fine-tuned foundation models is similar to or slightly worse than without finetuning. This may be due to the large number of parameters in the foundation models (e.g., BrainLM has 13 million parameters), which makes them more prone to overfitting when fine-tuned on the relatively small ADHD-Peking dataset (245 subjects versus over 40,000 in the pre-training datasets of baseline foundation models).

Table 8: Summary of additional target datasets, which are different imaging sites from the same cohort ADHD/ABIDE.

| Dataset | Total Subjects | Healthy Controls | Condition Subjects | Time Length |
|---|---|---|---|---|
| ADHD-NYU | 216 | 98 | 118 ADHD | 172 |
| ADHD-NI | 48 | 23 | 25 ADHD | 231 |
| ABIDE-NYU | 175 | 100 | 75 ASD | 176 |
| ABIDE-Leuven | 63 | 34 | 29 ASD | 246 |

Table 9: Performance on additional sites from ADHD and ABIDE dataset (complimentary for Section 5.1).

| Dataset | ADHD | | ABIDE | |
|---|---|---|---|---|
| Site | NYU | NI | NYU | Leuven |
| SVM | $0.6202 \pm 0.0662$ | $0.6887 \pm 0.0476$ | $0.6944 \pm 0.0755$ | $0.6672 \pm 0.0883$ |
| RF | $0.5605 \pm 0.0586$ | $0.6631 \pm 0.0591$ | $0.6785 \pm 0.0739$ | $0.6228 \pm 0.0579$ |
| ST-GCN | $0.5722 \pm 0.0694$ | $0.6013 \pm 0.0781$ | $0.6889 \pm 0.0621$ | $0.6786 \pm 0.0734$ |
| **MeTSK** | $\mathbf{0.6704 \pm 0.0782}$ | $\mathbf{0.8020 \pm 0.0914}$ | $\mathbf{0.7268 \pm 0.0687}$ | $\mathbf{0.7116 \pm 0.0502}$ |

Table 10: A comparison of MeTSK models pre-trained with different datasets as the target domain. Linear probing was performed for evaluation of models on the four clinical datasets.

| | | ADHD | | | ABIDE | | | Connectivity Features |
|---|---|---|---|---|---|---|---|---|
| | | Peking (proposed) | NYU | NI | UM | NYU | Leuven | |
| PTE | SVM | $\mathbf{0.6415 \pm 0.0312}$ | $0.5779 \pm 0.0469$ | $0.5589 \pm 0.0392$ | $0.5753 \pm 0.0459$ | $0.5661 \pm 0.0483$ | $0.5928 \pm 0.0475$ | $0.5697 \pm 0.0477$ |
| | RF | $0.5392 \pm 0.0553$ | $0.5395 \pm 0.0492$ | $0.5199 \pm 0.0508$ | $0.5156 \pm 0.0517$ | $0.5836 \pm 0.0558$ | $0.5783 \pm 0.0489$ | $0.5081 \pm 0.0612$ |
| | MLP | $0.5813 \pm 0.0504$ | $0.5283 \pm 0.0397$ | $0.5449 \pm 0.0517$ | $0.5038 \pm 0.0478$ | $0.5469 \pm 0.0527$ | $0.5592 \pm 0.0531$ | $0.5111 \pm 0.0402$ |
| OASIS-3 | SVM | $0.6407 \pm 0.0741$ | $0.6050 \pm 0.0812$ | $0.6255 \pm 0.0806$ | $0.6340 \pm 0.0889$ | $0.6567 \pm 0.1133$ | $0.6503 \pm 0.1097$ | $0.5541 \pm 0.0677$ |
| | RF | $\mathbf{0.6750 \pm 0.0753}$ | $0.5812 \pm 0.0711$ | $0.6718 \pm 0.0745$ | $0.6233 \pm 0.0769$ | $0.6066 \pm 0.1019$ | $0.5906 \pm 0.1083$ | $0.5329 \pm 0.0721$ |
| | MLP | $0.6034 \pm 0.0725$ | $0.5325 \pm 0.0791$ | $0.5563 \pm 0.0762$ | $0.5998 \pm 0.0824$ | $0.5277 \pm 0.0819$ | $0.5166 \pm 0.0729$ | $0.5231 \pm 0.0689$ |
| TaoWu | SVM | $\mathbf{0.6831 \pm 0.1431}$ | $0.5597 \pm 0.1419$ | $0.6807 \pm 0.1597$ | $0.6281 \pm 0.1766$ | $0.6073 \pm 0.1750$ | $0.6219 \pm 0.1476$ | $0.5725 \pm 0.1502$ |
| | RF | $0.6553 \pm 0.1701$ | $0.5375 \pm 0.1645$ | $0.5718 \pm 0.1821$ | $0.6328 \pm 0.1642$ | $0.5525 \pm 0.1725$ | $0.5810 \pm 0.1651$ | $0.6031 \pm 0.1631$ |
| | MLP | $0.6208 \pm 0.1582$ | $0.5625 \pm 0.1578$ | $0.5825 \pm 0.1577$ | $0.5750 \pm 0.1521$ | $0.5925 \pm 0.1593$ | $0.6050 \pm 0.1422$ | $0.5875 \pm 0.1631$ |
| Neurocon | SVM | $\mathbf{0.7529 \pm 0.1579}$ | $0.6794 \pm 0.1772$ | $0.6721 \pm 0.1842$ | $0.6874 \pm 0.1717$ | $0.6943 \pm 0.1635$ | $0.6760 \pm 0.1859$ | $0.6599 \pm 0.1807$ |
| | RF | $0.6230 \pm 0.1658$ | $0.5883 \pm 0.1807$ | $0.5756 \pm 0.1719$ | $0.5850 \pm 0.1826$ | $0.6084 \pm 0.1879$ | $0.5893 \pm 0.1551$ | $0.5427 \pm 0.1715$ |
| | MLP | $0.6100 \pm 0.1635$ | $0.5202 \pm 0.1473$ | $0.5478 \pm 0.1683$ | $0.5755 \pm 0.1709$ | $0.5497 \pm 0.1537$ | $0.5344 \pm 0.1681$ | $0.5313 \pm 0.1721$ |

## A.5 Experiment Results Using Additional Target Datasets

We present additional results by training the MeTSK model with different target datasets and subsequently evaluating its generalization performance on the same clinical tasks described in Section 5.2. Table 9 reports the knowledge transfer performance when evaluated on additional target datasets (other sites from the ADHD and ABIDE datasets). Table 10 shows the generalization performance on four unseen clinical datasets using MeTSK models pre-trained on different target datasets. The results consistently demonstrate performance improvements compared to using functional connectivity features, highlighting the robustness of the proposed representation learning strategy, which further validates the effectiveness of MeTSK in extracting transferable and generalizable representations across different clinical applications.

## A.6 Domain Similarity

To evaluate the transferability of learned features, we measure the distance between features extracted from different domains using Domain Similarity (Cui et al., 2018; Oh et al., 2022). We first compute the Earth Mover's Distance (EMD) (Yu & Herman, 2005), which is based on the solution to the Monge-Kantorovich problem (Rachev, 1985), to measure the cost of transferring features from the source to target domain. We define $\bar{X}_{\mathcal{S}} = \text{Flatten}(\frac{1}{N}\sum_{n=1}^{N}\tilde{X}_{\mathcal{S}})$, $\bar{X}_{\mathcal{T}} = \text{Flatten}(\frac{1}{N}\sum_{n=1}^{N}\tilde{X}_{\mathcal{T}})$ as the flattened vectors of the output graph features averaged over all subjects, and then define $B_s$ and $B_t$ as the set of bins in the histograms representing feature distribution in $\bar{X}_{\mathcal{S}}$ and $\bar{X}_{\mathcal{T}}$, respectively. A larger Domain Similarity indicates better

transferability from the source domain to the target domain because the amount of work needed to transform source features into target features is smaller. Domain similarity (DS) is defined as:

$$\text{DS} = \exp\left(-\gamma \, \text{EMD}(\bar{X}_{\mathcal{S}}, \bar{X}_{\mathcal{T}})\right) \tag{7}$$

$$\text{EMD}(\bar{X}_{\mathcal{S}}, \bar{X}_{\mathcal{T}}) = \frac{\sum_{i=1}^{|B_s|} \sum_{j=1}^{|B_t|} f_{i,j} d_{i,j}}{\sum_{i=1}^{|B_s|} \sum_{j=1}^{|B_t|} f_{i,j}},$$

$$s.t. \quad f_{ij} \geq 0,$$

$$\sum_{j=1}^{|B_t|} f_{ij} \leq \frac{|\bar{X}_{\mathcal{S}} \in B_s(i)|}{|\bar{X}_{\mathcal{S}}|}, \tag{8}$$

$$\sum_{i=1}^{|B_s|} f_{ij} \leq \frac{|\bar{X}_{\mathcal{T}} \in B_t(j)|}{|\bar{X}_{\mathcal{T}}|},$$

$$\sum_{i=1}^{|B_s|} \sum_{j=1}^{|B_t|} f_{ij} = 1$$

where $B_s(i)$ is the i-th bin of the histogram and $|B_s|$ is the total number of bins, $|\bar{X}_{\mathcal{S}} \in B_s(i)|$ is the number of features in $B_s(i)$, $|\bar{X}_{\mathcal{S}}|$ is the total number of features, $d_{i,j}$ is the Euclidean distance between the averaged features in $B_s(i)$ and $B_t(j)$, $f_{i,j}$ is the optimal flow for transforming $B_s(i)$ into $B_t(j)$ that minimizes the EMD. Following the setting in (Cui et al., 2018), we set $\gamma = 0.01$.

### A.6.1 Experiments Using Domain Similarity

To further investigate the transferability enabled by MeTSK, domain similarity was computed to evaluate the knowledge transfer from control data (source) to clinical data (target) as well as from the training set to the testing set of target data. We conducted domain similarity analysis on both ADHD-Peking and ABIDE-UM datasets to further validate the robustness and versatility of MeTSK. Fig. 6 illustrates that the self-supervised source features have a higher similarity with the target features, indicating better inter-domain transferability and thus improved performance on the target classification task. Moreover, compared to the baseline, both intra-ADHD-class/intra-ASD-class and intra-TDC-class domain similarities between the training and testing sets of ADHD/ABIDE data are increased by MeL. This enhancement provides evidence to explain the improved classification performance on training with only target data achieved by meta-learning. By applying meta-learning, not only the inter-domain generalization of features is boosted, but also the effect of heterogeneous data within the same domain is alleviated.

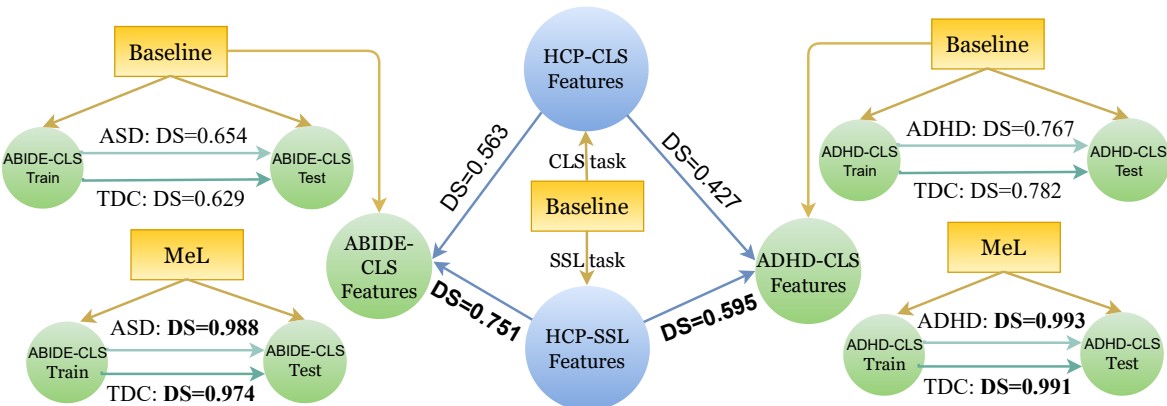

Figure 6: A comparison of the domain similarity between HCP self-supervised features (HCP-SSL, from Baseline ST-GCN trained on HCP data with a self-supervised task) and ADHD/ASD classification features (ADHD-CLS, ABIDE-CLS, from Baseline trained using all ADHD/ABIDE data), the domain similarity between HCP sex classification features (HCP-CLS, from Baseline trained on HCP data with a sex classification task) and ADHD-CLS/ABIDE-CLS, the intra-class (ADHD; TDC and ASD; TDC) domain similarities between training and testing set of ADHD/ASD data from Baseline and MeL (a meta-learning model trained only on target data), respectively.

