# OpenReview forum: "Generalizable Representation Learning for fMRI-based Neurological Disorder Identification"
_TMLR — Accepted by TMLR_

### Review · Reviewer_naZS · 2025-02-20

**Summary Of Contributions:**

The authors consider the task of training classifiers to distinguish between neurological disorder patients and healthy controls, using fMRI scans.
They propose a method to overcome the scarcity of patient data by training a feature extractor with self-supervised learning on scans from healthy controls, which are more easily available.
This approach yields better results than several baselines on 4 independent datasets.

The input features are connectivity matrices between regions of the AAL atlas computed from resting-state fMRI.
The self-supervised task is contrastive learning; 2 (time series of) connectomes should be more similar if they are from the same subject than if they are from 2 different subjects.
The whole model is trained by optimizing both the performance on this self-supervised task and on the target classification task.
This is done by alternating between 2 steps.
In the "inner" loop, the feature extraction parameters are frozen and a "target head" is trained to perform the classification task, using the extracted features.
Then in the "outer" loop, the target head is frozen, and the feature extraction parameters are updated to optimize a combination of the performance on the self-supervised contrastive learning task and of the performance of the (frozen) target head applied to the extracted features (on an unseen sample, ie a sample that was not used to train the target head).

HCP is used as the source dataset (for the self-supervised training), and ADHD and ABIDE are used as target datasets (to which the target head is applied during training).

Once trained, the model is evaluated on four independent datasets: MagNeTs, OASIS, TaoWu, Neurocon.
The evaluation is done by using the extracted features as inputs to a linear classifier and measuring its area under the ROC curve with 5-fold cross-validation.
The proposed method outperforms several baselines in this setting.
One of the baselines is the more traditional approach of training a classifier on the flattened connectivity matrix extracted from fMRI time series (ie no pre-trained feature extraction).

**Audience:**

Yes

**Claims And Evidence:**

Yes

**Requested Changes:**

- Could the authors clarify what statistical test was performed to compare models
- Could the authors provide more details about the fMRI preprocessing, construction of connectome time series that are input to the feature extractor, and justify the choice of AAL atlas and Pearson correlation.
- Only a small subsample of the OASIS evaluation dataset is used; could the authors explain why? The small-data regime could still be simulated by using a small training set, while using the whole dataset for testing and thus producing more accurate evaluation scores.
- Could the authors include information and a discussion about the computational cost of the proposed method and the baselines


typo in the introduction:

- "simply train a model" -> "simply training"

**Strengths And Weaknesses:**

# strengths

The scarcity of clinical data is an important challenge.
Leveraging more abundant healthy control data is a sound idea to address it, and has received much attention over the years, but how to do this remains an open question and the authors contribute to that search by applying methods from the recent machine-learning literature.

Overall the paper is well written and illustrated, the diagrams help understand the proposed method.

The model yields better predictive performance than several baselines.

Several independent datasets are used for evaluation.

## weaknesses

If I understand correctly, the authors perform a Student t-test on cross-validation scores (with overlapping folds). Those scores are not independent, violating a crucial hypothesis of the test and invalidating its results. If I misunderstood, please provide more details on what test was performed. And if I did understand and the t-test was done on non-iid samples please remove the claims of statistical significance.
As several independent datasets are available, perhaps this could be leveraged for a global comparison of methods instead:

Demšar, J. (2006). Statistical comparisons of classifiers over multiple data sets. The Journal of Machine learning research, 7, 1-30.

More details should be provided about the fMRI preprocessing and construction of the connectivity matrices.

Pearson correlation is probably not the best way to build the connectome, at least for the baseline classifiers that use these features directly. See for example

Stephen M. Smith, Karla L. Miller, Gholamreza Salimi-Khorshidi, Matthew Webster, Christian F. Beckmann, Thomas E. Nichols, Joseph D. Ramsey, and Mark W. Woolrich. Network modelling methods for FMRI. NeuroImage,

Gaël Varoquaux and R. Cameron Craddock. Learning and comparing functional connectomes across subjects. NeuroImage, 80:405–415, 2013. Mapping the Connectome.

Moreover, a set of learn functional regions (eg the DiFuMO atlas) may yield better performance than the AAL atlas regions. Therefore the baseline might be a bit weaker than the state of the art of "simple" methods.

The clinical relevance of the evaluation tasks could be discussed more. Classifying Alzheimer or PD patients vs controls can usually be done more reliably with structural images; the main motivation to tackle such tasks with fMRI is often the hope to provide insights into the disease or its evolution by inspecting the model but this could be hindered by the increased complexity of the feature extraction process.

---

> ### Author Response · Authors · 2025-04-04
> **Authors Response to Reviews**
>
> We sincerely thank the reviewer for providing insightful and constructive comments, which have greatly contributed to improving the clarity and quality of our paper. Please find our detailed responses below.
>
> ---
>
> ### **1. Use of Student's t-test on Cross-Validation Scores**
>
> **Response:**
> We thank the reviewer for this important point. We agree that applying a standard paired Student's t-test on k-fold cross-validation scores can violate the independence assumptions due to overlapping training sets during cross-validation.
> As mentioned in the reference "Demšar, J. (2006). Statistical comparisons of classifiers over multiple data sets. The Journal of Machine learning research, 7, 1-30." ,  Wilcoxon signed-rank test is recommended.
> We have revised our manuscript to remove claims of statistical significance derived from the paired t-test and added a discussion in Section 5.2.2 about the p-values computed using Wilcoxon signed-rank test. The lowest p-value we obtained from the Wilcoxon signed-rank test was 0.0625. The method pairs yielding these lowest p-values were consistent with those identified as most significant in the original paired Student’s t-tests as shown in Table 2. Using more cross-validation folds for larger datasets in future work could yield more granular and reliable statistical significance estimates.  Additional p-values are also presented in Appendix A.2 Table 6.
> We greatly appreciate the reviewer bringing this to our attention, allowing us to significantly improve the rigor of our manuscript.
>
> ---
>
> ### **2. Details on fMRI Preprocessing, Construction of Connectome Time Serie, Atlas and Connectivity Measure Choice**
>
> **Response:**
> Thank you for the suggestion.
> We have added more details about the fMRI preprocessing steps and the construction of functional connectivity measures in Section 4 of the revised manuscript. For most of the datasets, we used publicly available preprocessed versions to ensure consistency with standard pipelines and enhance reproducibility.
>
> We chose the AAL atlas due to its wide adoption in neurological disorder studies. Notably, large-scale public datasets such as the ADHD-200 Consortium and ABIDE use the AAL atlas as part of their standard preprocessing pipelines. Additionally, the AAL atlas is publicly available, easy to implement across datasets, and widely validated in the literature, enhancing the reproducibility and general applicability of our approach.
>
> Similarly, we adopt Pearson correlation as the measure of functional connectivity because it remains the most commonly used method in the field. While other alternatives (e.g., partial correlation) exist, Pearson correlation provides a direct and computationally efficient means to estimate pairwise functional connectivity. Exploring alternative measures is an interesting direction for future work, but it is beyond the primary focus of this study.
>
> We also emphasize that our method is not inherently tied to a specific atlas or connectivity measure. We have released the source code to facilitate experimentation with different choices, such as alternative atlases or connectivity metrics. We have added the references provided by the reviewer and included this as possible future work in the Discussion part of the paper: Section 7.
>
> ---
>
> ### **3. Use of a Subsample from OASIS-3**
>
> **Response:**
> To maintain consistent evaluation protocols across all datasets, including those with naturally limited sample sizes, we applied a 5-fold cross-validation setup using a fixed subsample of the OASIS dataset.
> We chose cross-validation for two reasons: (1) it provides a balanced estimate of generalization performance across multiple folds, and (2) it ensures that the evaluation setup is consistent across datasets, especially in scenarios where there is no additional held-out test set.
>
> The OASIS-3 dataset contains a larger number of control subjects than AD subjects. In our study, we selected a balanced subsample (42 AD vs. 42 Control) from the entire dataset to ensure a fair binary classification setup. We intend to avoid class imbalance effects and to maintain consistency with our focus on evaluating model generalization in low-data regimes.
> We add this discussion in Section 7 of the revised manuscript.
>
> ---
>
> ### **4. Computational Cost of MeTSK and Baselines**
>
> **Response:**
>
> Thanks for the suggestion.
> We have added this in the Implementation Details: Section 5.3 of the revised manuscript.
>
> ---
>
> ### **5. Clinical Relevance of the Evaluation Tasks**
>
> **Response:**
>
> We appreciate the reviewer’s insightful comment. Indeed, structural imaging is commonly used in clinical practice for diagnosing disorders such as Alzheimer’s Disease and Parkinson’s Disease. fMRI offers complementary information by capturing dynamic patterns of brain activity.
> We plan to further explore the interpretability of the learned representations in future work, especially for AD and PD.
>
> ---

---

> > ### Comment · Reviewer_naZS · 2025-04-15
> >
> > Thank you for addressing my comments. I am a bit unclear about how the Wilcoxon signed-rank test was used -- the reference describes a way to do comparison across multiple datasets, but the Table 2 gives the impression that a different statistic was computed for each dataset and method pair, rather than across datasets? also note increasing the number of cross-validation folds would not add new data or reduce the overlap, so adding new datasets would be the way to increase the significance of the comparison. Could the authors provide more details about the test that was conducted? note that it is perfectly possible and very common to present cross-validation results without claims of statistical significance or p-values, and that might be a good option here especially since the p-value is rather high
> >
> > I think all my other comments have been answered

---

> > > ### Author Response · Authors · 2025-04-23
> > > **Thanks for Your Comments**
> > >
> > > We greatly appreciate the reviewer's insightful comments. The reviewer's understanding is correct—our use of the Wilcoxon signed-rank test involved separate evaluations on each dataset's 5-fold results across methods. As suggested, we will remove all claims of statistical significance and present the cross-validation results in Table 2 without p-values. The previously included p-values were intended as illustrative rather than formal evidence.

---

### Review · Reviewer_hBfH · 2025-03-10

**Summary Of Contributions:**

This paper addresses the poor generalization of deep learning models in neurological disorder classification due to the scarcity and heterogeneity of clinical fMRI data, while healthy control data is abundant. To tackle this problem, they propose MeTSK, a representation learning strategy that integrates self-supervised learning with meta-learning. The problem is formulated as a domain adaptation task, where the goal is to generalize from a source domain (healthy controls) to a target domain (clinical data). Their algorithm consists of two steps: (1) contrastive self-supervised learning applied to healthy control data to learn generalizable representations, and (2) bi-level meta-learning used to optimize feature transfer from the source to the target domain. The inner loop adapts the model to the clinical domain, while the outer loop ensures that the learned features remain generalizable.
The authors evaluated their method on four clinical fMRI datasets, including ADHD, Autism Spectrum Disorder (ASD), Post-Traumatic Epilepsy (PTE), Alzheimer’s Disease (AD), and Parkinson’s Disease (PD). They compared MeTSK to multiple baseline methods, including fine-tuning, multi-task learning, and foundation models. Results showed MeTSK outperforms other approaches in classifying neurological disorders from limited clinical data.
In their ablation study, they investigated the individual contributions of meta-learning and self-supervised learning. They trained a version of the model using only meta-learning without self-supervised learning and another using self-supervised learning without meta-learning. They also test the impact of using different source tasks (e.g., sex classification instead of self-supervised learning). Their results confirmed that both self-supervised learning and meta-learning contribute to performance, and transferring from task-agnostic self-supervised features is more effective than transferring from sex-based classification.

**Audience:**

Yes

**Claims And Evidence:**

Yes

**Requested Changes:**

### Major comments:

- MeTSK simultaneously optimizes self-supervised InfoNCE loss and supervised cross-entropy loss, which differ in type and scale. A formal convergence guarantee or empirical analysis of training stability is required. Clarify how convergence is monitored and how updates in Step 2-\theta_t and Step 3-\phi do not negatively interfere with each other.

- The generalization performance of MeTSK may stem from either the method or the pre-training data. To ensure a fair comparison, it is required to either fine-tune baseline foundation models on similar pre-training data, or align MeTSK's pre-training dataset with the baseline models. Without this, it remains unclear whether the improvement comes from methodology or data differences.

- Providing source code is required to validate results and ensure reproducibility. Also implementation details are totally missed in the paper.

### Minor comments:

- Clarification of ADHD-All vs. ADHD-Peking performance in Figure 3, PTE case

- Improvements in Text and Captions:
* Some captions are incomplete and should be revised for clarity.
* Table 3: Clearly define the baseline model in both the body-text and caption.
* Section 5.1: Specify what classification method is used for MeTSK (or provide a reference if mentioned earlier).
* Section 4.5 (End): PD or AD?

**Strengths And Weaknesses:**

### Strengths:

- The proposed method is interesting and practical and shows promising effectiveness. Especially, the integration of self-supervised learning and meta-learning is well-motivated and addresses the challenge of adapting fMRI representations to clinical data.
- The evaluation is thorough, with comparisons across multiple baselines and datasets.
- The paper is well-written and easy to follow.

-----

### Weaknesses:

- MeTSK contains a multi-objective loss that simultaneously optimizes two distinct functions—self-supervised InfoNCE loss for the source domain (healthy controls) and supervised cross-entropy loss for the target domain (clinical data). Given that these losses differ in both type and scale, this raises concerns about training stability and convergence. How do the authors monitor convergence to ensure balanced optimization of both objectives? Additionally, when updating \( \theta_t \) in Step 2, how is it ensured that this update does not adversely affect the optimization process in Step 3? Similarly, when updating \( \phi \) in Step 3, how do the authors prevent it from negatively impacting the update in Step 2? Providing a formal convergence guarantee or an empirical analysis of training stability is required for the claims in the paper.
- As the authors also noted, "due to the domain gap between their pre-training data and the clinical datasets used here, the foundation models may require additional fine-tuning to further improve their performance.", it raises concerns regarding the generalizability evaluation of the proposed method. Specifically, the observed superior generalization performance of MeTSK could be attributed either to the method itself or to the data used for pre-training. To ensure a fair comparison, either (1) the baseline foundation models should be fine-tuned on similar pre-training data, or (2) the pre-training dataset for MeTSK should align with the data used in the baselines. Without this, it remains unclear whether the advantage comes from the methodology or the pre-training data.
- In Figure 3, why does training on ADHD-All perform worse than ADHD-Peking alone in the PTE case? Is this due to the domain shift between different sites?
- It is important to provide source code for reproducibility of the results.
- Text at some parts should be improved. For example, some captions are incomplete and should be revised to clearly convey their message and match the titles. In Table 3, the definition of the baseline model should be explicitly clarified both in the text and caption. In Section 5.1, it is not mentioned what classification is used for MeTSK. If it is mentioned before, please remind the reader. At the end of Section 4.5, is it PD or AD?

---

> ### Author Response · Authors · 2025-04-04
> **Author's Response to Reviews**
>
> We sincerely thank the review for providing comprehensive and helpful comments. Please find our detailed responses below.
>
> ### **1. What classification method is MeTSK using?**
>
> **Response:**
>
> It is a linear layer plus sigmoid activation to perform binary classification. The loss is binary cross entropy. We have added the detailed explanation in Section 5.3.
>
> ---
>
> ### **2. Typo Correction in Section 4.5; Some captions are incomplete and should be revised for clarity. Table 3 baseline Clarification.**
>
> **Response:**
>
> Thank you so much for spotting this.
> The typo has been corrected from PD to AD in Section 4.5.
>
> We have revised and completed all captions for improved clarity.
>
> We have clarified the baseline model in Table 3.
>
> ---
>
> ### **5. Performance drop of ADHD-All in Figure 3 (PTE dataset)**
>
> **Response:**
>
> We have added discussion on this in Section 6.1.  " Shen, J. H., Raji, I. D., Chen, I. Y. (2024). The data addition dilemma. arXiv preprint arXiv:2408.04154" highlights that adding more data from heterogeneous sources can in some cases degrade model performance due to domain shifts.
>
> ---
>
> ### **6.  MeTSK simultaneously optimizes self-supervised InfoNCE loss and supervised cross-entropy loss, which differ in type and scale. A formal convergence guarantee or empirical analysis of training stability is required. Clarify how convergence is monitored and how updates in Step 2-$\theta_t$ and $Step 3-\phi$ do not negatively interfere with each other.**
>
> **Response:**
> We thank the reviewer for this important observation. We provide the following clarification and have added a figure in Appendix A.3 illustrating the evolution of both the self-supervised loss and the classification loss (cross-entropy loss) over training iterations.
> It is true that the self-supervised InfoNCE loss and the cross-entropy loss have different scales. We used a hyper-parameter $\lambda$ to balance the scales of the two losses (introduced in Section 3.2 and 5.3).
>
> We understand that the reviewer's concern is that the target head is fitted to the meta-training set in Step 2, and updating only the feature extractor in Step 3 may not be sufficient to reduce the classification loss on the meta-validation set. To address this, we track both the self-supervised InfoNCE loss and the cross-entropy loss computed from the frozen target head’s predictions in Step 3. During training, we observe that the InfoNCE loss decreases well, while the cross-entropy loss also shows a decreasing trend, though with some fluctuations.
>
> The reviewer is also concerned that the updates in Step 3 — which optimize the feature extractor using the self-supervised loss and the classification loss on the meta-validation set — could interfere with the target head updates in Step 2 during the next iteration. In our framework, the target head is randomly initialized in every training iteration, and both the meta-training and meta-validation sets are randomly re-sampled. This setup encourages the model to generalize and quickly adapt to new data.
> We observe that the classification loss in Step 2 consistently decreases to a stable range after the inner loop updates in each training iteration. This indicates that the target head update process in Step 2 is not negatively affected by the updates made in Step 3.
>
> ---
>
> ### **7. The generalization performance of MeTSK may stem from either the method or the pre-training data. To ensure a fair comparison, it is required to either fine-tune baseline foundation models on similar pre-training data, or align MeTSK's pre-training dataset with the baseline models. Without this, it remains unclear whether the improvement comes from methodology or data differences.**
>
> **Response:**
> We appreciate the reviewer’s suggestion regarding fine-tuning the foundation models on data comparable to that used by our method.
> We have added the linear probing results of fine-tuned foundation models alongside the results before fine-tuning in Appendix A.4, Table 7. To ensure a fair comparison, we fine-tuned the baseline foundation models on the same dataset—HCP combined with ADHD-Peking—that was used for training MeTSK. Notably, the pre-training datasets for these foundation models already included the HCP data, so aligning the training data required only fine-tuning on the ADHD-Peking dataset.
> Additionally, the SSL model in Table 2 trained on the same datasets as MeTSK using only a contrastive self-supervised loss achieved worse performance than our MeTSK model.
>
> ---
>
> ### **8. Source code availability and implementation details**
>
> **Response:**
> Thank you for the suggestion. We will release our code publicly upon publication of the paper. In the meantime, we have uploaded the source code as the supplementary materials. Due to page limits, the implementation details were previously included in the Appendix. We have now moved them to Section 5.3 of the main paper and added more information.
>
> ---

---

> > ### Comment · Reviewer_hBfH · 2025-04-20
> > **Response to the rebuttal**
> >
> > I thank the reviewers for their responses to my comments and for addressing most of the points I raised.

---

### Review · Reviewer_ioAF · 2025-03-21

**Summary Of Contributions:**

The work presents a deep learning approach for more robust medical predictions (like Alzheimers or ADHD classification) from fMRI. Bi-level Optimization (similar to MAML) learns generic features and task-specific heads are then used to classify fMRI from different datasets on top of them. Additionally, a self-supervised contrastive loss is used to further train the features to be more robust and also allow learning from unlabeled data. As a feature extractor for fMRI,a Spatio-temporal Graph Convolutional Network is used. The model is evaluated on unseen clinical datasets with different clinical tasks.  SVMs, Random Forests and MLPs are used as classifiers on top of the learned features. Results show that the proposed approach yields features which lead to better accuracies than features learned by some other approaches.

**Audience:**

Yes

**Claims And Evidence:**

No

**Requested Changes:**

For downstream evaluation, please also add :

* result of linear classifier trained as it was trained during meta training stage (e.g., instead of linear SVM)
* results of ST-GCN trained from scratch
* results from published literature wherever available for the datasets

**Strengths And Weaknesses:**

The paper has a clear motivation and a fairly straightforward approach to tackle the proposed problem.

Writing is  understandable and figures seem clean to me.

I find the proposed evaluation a bit strange with SVM/RF/MLP, it would seem more straightforward to always just try actual linear classifier? especially if you call it linear probing and train for it to be linearly seperable in the meta-training stage. Also I would have also expected to see performance of ST-GCN trained from scratch on each dataset, as well as results directly taken from the published literature.

---

> ### Author Response · Authors · 2025-04-04
> **Response to Reviews: Clarifications, Additional Results, and Revisions**
>
> Thanks for your time and effort in reviewing our manuscript. We appreciate the reviewer for providing constructive and insightful comments, which have helped us improve both the clarity and rigor of our paper. Our point-by-point responses to your comments are provided below.
>
> ### **1. Result of linear classifier trained as it was trained during meta training stage (e.g., instead of linear SVM)**
>
> **Response:**
> We thank the reviewer for raising this thoughtful point. In the representation learning and foundation model literature, linear probing is widely used to evaluate the linear separability of learned representations by training simple classifiers—typically logistic regression or linear SVM—on frozen features. For instance, the BrainLM foundation model (“BrainLM: A Foundation Model for Brain Activity Recording” by Ortega Caro et al.) uses linear SVM for representation evaluation.
> In our work, we originally adopted a linear SVM, which has a linear decision boundary and aligns with this standard evaluation protocol. Linear probing is also widely adopted in the literature, where logistic regression or a single linear layer followed by softmax is commonly used.
>
> To improve clarity and consistency with standard linear probing protocols, we replaced the previously used MLP classifier with logistic regression (a linear layer followed by a sigmoid activation) and updated the results in Table 2 and Figure 4. We also added clarifying statements about our linear probing setup in Section 3.4 (page 6) of the revised manuscript.
>
> ---
> ### **2. Results of ST-GCN trained from scratch**
>
> **Response:**
> We previously reported the results of ST-GCN trained from scratch in Appendix A.1, Table 4, alongside other baseline models from published literature. Please also the the response for the next comment.
>
> ---
>
> ### **3. Results from published literature wherever available for the datasets**
>
> **Response:**
> We thank the reviewer for this helpful suggestion. Due to methodological and experiment setting differences across previous works (e.g., varying cross-validation splits, fMRI atlas choices, and number of training samples used), direct numerical comparisons should be interpreted cautiously. Nonetheless, we include these published results from previous works to contextualize our results within the broader literature. We summarized this comparison in Appendix A.1, Table 5.
>
> Additionally, in Appendix A.1, Table 4 (included in the original manuscript), we report results obtained by implementing several published models for fMRI-based neurological disorder classification on the four clinical datasets used in our study.
>
> Unlike previous works that often leverage larger datasets, our experimental design simulates real-world clinical scenarios with limited training data. This allows us to assess the robustness and generalization capability of MeTSK under data-scarce conditions—settings that are common in clinical practice.
> Since our primary goal is to learn representations that generalize across domains, we focus on comparisons with published foundation models and self-supervised learning (SSL) approaches in Table 2 of the main paper.

---

### Review · Reviewer_qJ4R · 2025-03-26

**Summary Of Contributions:**

This paper aims to improve the generalizability of deep learning models of neurological disorder classification in fMRI data. To mitigate the challenge of scarcity in clinical data, the authors proposed a meta-learning and self-supervised learning framework to learn generalizable representations from fMRI data of patients and healthy subjects. The authors showed that their method outperforms baseline models and previously published methods in both direct training and transfer learning setups. Moreover, they compared their method against two publicly available fMRI foundation models on four different clinical fMRI datasets. They showed that the model achieved better performance in disease vs healthy control classification.

**Audience:**

Yes

**Broader Impact Concerns:**

I do not believe this work requires a broader impact statement.

**Claims And Evidence:**

Yes

**Requested Changes:**

- Rather than treating the 4 clinical datasets (i.e. results in Table 2) as 4 separate binary disorder/disease vs control classifications, have the authors experimented with treating the 4 datasets as a 5-class classification instead? It would be interesting to see how MeTSK performs against the two fMRI foundation models since all healthy control data from the 4 datasets would be combined into a single class and models pre-trained on large amounts of data might be beneficial.
- Can the authors show the statistical test result of each pair of bars in Figure 4? The error bars show there is a decent amount of overlap against the baseline connectivity features (i.e. yellow bars) too.
- Following the previous comment, the authors mentioned there is no significant difference in performance between the model trained on all data (`ADHD-All`) vs data from a single site (`ADHD-Peking`). Following the recent approach of foundation models, one would expect the model to improve by increasing the training sample size. Figure 4 shows that the performance of MeTSK does not necessarily scale with training data. I wonder if the authors can comment on whether or not this is a desirable behaviour.
- Can the authors briefly explain how the hyperparameters were selected? Are the hyperparameters presented in Appendix A.3 selected via an optimization process (e.g. grid search, bayesian search, etc.) or are they handpicked?
- Minor typos and `\citep` vs `\citet`. e.g. Page 2 According to Kumar el. al., (2022), Page 3 end of the second paragraph, Page 9 first second of the third paragraph, etc.

**Strengths And Weaknesses:**

Strengths
- The method evaluation analysis is comprehensive. The authors have included a wide range of baseline models, datasets, as well as training configurations (direct training vs fine-tuning vs self-supervised).
- Overall, the paper is well-written and easy to follow.

Weaknesses
- I don’t have any major concerns with the paper but I have some minor questions and suggestions, please see below.

---

> ### Author Response · Authors · 2025-04-04
> **Response to Reviewer Comments: Clarifications and Revisions**
>
> We sincerely thank the reviewer for their constructive feedback. We have carefully addressed your suggestions and questions, and we have incorporated revisions accordingly. Please find our detailed responses below, and we welcome any further feedback and discussion.
>
> ---
>
> ### **1. Multi-class Classification across Datasets**
>
> **Response:**
> We thank the reviewer for this interesting suggestion.  We agree that a multi-class setup could offer insights. However, implementing such a setup introduces non-trivial challenges due to inter-dataset discrepancies. The datasets are collected from different imaging sites and scanners (e.g., the PD-Taowu dataset was acquired using a 3T MRI scanner, whereas PD-Neurocon used a 1.5T scanner), and their temporal resolution, demographics, scanners, and protocols are all different, leading to potential domain shifts. Additionally, the control group in the PTE dataset consists of subjects with traumatic brain injury but without post-traumatic epilepsy or other neurological conditions—different from healthy controls in other datasets.
>
> Combining these control groups into a single class could introduce distribution shifts, potentially affecting the evaluation. Given these concerns, we chose to evaluate the datasets independently in a binary classification setting.
> Nevertheless, we acknowledge the value of the proposed multi-class experiment and added this discussion in Section 7 of the revised manuscript as potential future work.
>
> ---
>
> ### **2. Statistical Test Results for Figure 4**
>
> > Can the authors show the statistical test result of each pair of bars in Figure 4? The error bars show there is a decent amount of overlap against the baseline connectivity features (i.e. yellow bars) too.
>
> **Response:**
> We have modified the paper and now clarify in Section 5.2.2 that instead of using a paired Student's t-test, the Wilcoxon signed-rank test is more appropriate. We provided the p-values computed using the Wilcoxon signed-rank test between the results from connectivity features, MeTSK-ADHD-All, and MeTSK-ADHD-Peking in Appendix A.2 Table 6.
>
> ---
>
> ### **3. Performance Scaling with Data Size**
>
> > Following the previous comment, the authors mentioned there is no significant difference in performance between the model trained on all data (ADHD-All) vs data from a single site (ADHD-Peking). Following the recent approach of foundation models, one would expect the model to improve by increasing the training sample size. Figure 4 shows that the performance of MeTSK does not necessarily scale with training data. I wonder if the authors can comment on whether or not this is a desirable behaviour.
>
> **Response:**
> We thank the reviewer for this observation. We have added discussion on this in Section 6.1. While it is generally expected that model performance improves with more training data, this assumption may not hold when the additional data introduces substantial domain shifts. In our case, the ADHD-All dataset includes fMRI scans from multiple imaging sites with different acquisition protocols, which can introduce distribution shifts.
> This phenomenon has been studied in prior work, "Shen, J. H., Raji, I. D., Chen, I. Y. (2024). The data addition dilemma. arXiv preprint arXiv:2408.04154", which highlights that adding more data from heterogeneous sources can in some cases degrade model performance due to domain shifts.
> We have now added this explanation and the reference in Section 6.1.
>
> ---
>
> ### **4. Hyperparameter Selection Process**
>
> > Can the authors briefly explain how the hyperparameters were selected? Are the hyperparameters presented in Appendix A.3 selected via an optimization process (e.g. grid search, bayesian search, etc.) or are they handpicked?
>
> **Response:**
> The hyper-parameters for the baseline ST-GCN model and the contrastive SSL loss follow the settings reported in their respective original papers. Other hyper-parameters in MeTSK were selected empirically by observing the convergence of the training loss. The hyper-parameters in SVM and RF during linear probing were selected via grid search using a nested cross-validation. We have clarified this and included the information in Section 5.3: Implementation Details.
>
> ---
>
> ### **5. Minor Typos and Citation Style**
>
> **Response:**
> Thank you so much for pointing these out. We have corrected the citation formatting throughout the manuscript.

---

> > ### Comment · Reviewer_qJ4R · 2025-04-23
> >
> > I thank the authors for their detailed response, which addressed my main concerns and questions.

---

### Decision · Action_Editor_EVye · 2025-05-12

**Recommendation:** Accept with minor revision

**Comment:**

I believe the latest submission has the changes made to account for reviewers' comments in blue. After the color is harmonized with the rest of the paper's text, the paper is ready.

**Audience:**

The paper would be interesting for TMLR readers and this statement is supported by all reviewers.

**Claims And Evidence:**

The claims are supported by evidence and as pointed out by most reviewers the paper provides a thorough and comprehensive evaluation.

---

> ### Author Response · Authors · 2025-05-27
>
> We sincerely thank the Reviewers and Action Editor for their valuable input and suggestions.